# The Challenges of Machine Learning: A Critical Review

**Enrico Barbierato** *,† and **Alice Gatti** †

Department of Mathematics and Physics, Catholic University of the Sacred Heart, 25133 Brescia, Italy; alice.gatti@unicatt.it

\* Correspondence: enrico.barbierato@unicatt.it

† These authors contributed equally to this work.

**Abstract:** The concept of learning has multiple interpretations, ranging from acquiring knowledge or skills to constructing meaning and social development. Machine Learning (ML) is considered a branch of Artificial Intelligence (AI) and develops algorithms that can learn from data and generalize their judgment to new observations by exploiting primarily statistical methods. The new millennium has seen the proliferation of Artificial Neural Networks (ANNs), a formalism able to reach extraordinary achievements in complex problems such as computer vision and natural language recognition. In particular, designers claim that this formalism has a strong resemblance to the way the biological neurons operate. This work argues that although ML has a mathematical/statistical foundation, it cannot be strictly regarded as a science, at least from a methodological perspective. The main reason is that ML algorithms have notable prediction power although they cannot necessarily provide a causal explanation about the achieved predictions. For example, an ANN could be trained on a large dataset of consumer financial information to predict creditworthiness. The model takes into account various factors like income, credit history, debt, spending patterns, and more. It then outputs a credit score or a decision on credit approval. However, the complex and multi-layered nature of the neural network makes it almost impossible to understand which specific factors or combinations of factors the model is using to arrive at its decision. This lack of transparency can be problematic, especially if the model denies credit and the applicant wants to know the specific reasons for the denial. The model's "black box" nature means it cannot provide a clear explanation or breakdown of how it weighed the various factors in its decision-making process. Secondly, this work rejects the belief that a machine can simply learn from data, either in supervised or unsupervised mode, just by applying statistical methods. The process of learning is much more complex, as it requires the full comprehension of a learned ability or skill. In this sense, further ML advancements, such as reinforcement learning and imitation learning denote encouraging similarities to similar cognitive skills used in human learning.

**Keywords:** machine learning; scientific method; imitation learning; mirror neurons





## 1. Introduction

The notion of learning is far from sharing a unique interpretation as the scientific literature presents different perspectives, ranging from pedagogic to philosophic approaches, even involving sociology or hard sciences. For example, according to Bloom [1], learning is the acquisition of knowledge or skills ("Learning is a relatively permanent change in behavior potentiality that occurs as a result of reinforced practice or experience"), while according to Jonassen [2], learning is the construction of meaning ("Learning is the process whereby individuals construct meanings based on their experiences."). Authors such as Vygotsky [3] consider learning as a social phenomenon, which originates from dynamic transactions between an individual and the surrounding environment. Similarly, Piaget [4] relates learning to social development, besides requiring information acquisition. As the process necessitates an intense social exchange, Jarvis [5] claims that it is not related to time or places but is an ongoing activity.

ML seeks to exploit supervised or unsupervised algorithms that can learn a specific task (for example, classification or clustering) from a dataset. The former technique requires a dataset including the specification of examples of a target variable under study; the remaining variables are called predictors. The latter technique involves, for example, dataset clustering, which returns in output two or more data clusters, whose points are grouped according to spatial distance and, ultimately, by similarity. Other cases of unsupervised learning include self-organizing maps (SOMs), principal component analysis (PCA), and so forth. A training algorithm takes as an input a dataset of known observations, which is partitioned into "train" (TR) and "test" (TS) subsets (80% and 20% of the initial dataset). The algorithm is then trained on TR until it can achieve the task with acceptable performance. In the second phase, the prediction ability of the algorithm is tested against the "test" dataset. ML researchers claim that an algorithm has learned a task when it can generalize its judgment when considering new observations that were not part of the original dataset. More formally, determining whether an ML model has "learned" or not depends on the specific context and the goals of the model. However, some general criteria can be considered to assess whether a model has successfully learned, as per Table 1.

**Table 1.** ML model learning benchmarks.

| Benchmark | Description |
| --- | --- |
| Accuracy | The model should be able to make accurate predictions or classifications on unseen data. This means that the model should generalize well beyond the training data it was exposed to. |
| Generalizability | The model should not be overly specific to the training data and should be able to perform well on data from different sources or with different distributions. |
| Robustness | The model should be resistant to noise and outliers in the data and should not be easily fooled by adversarial examples. |
| Interpretability | The model should be understandable and explainable, allowing us to understand how it makes its decisions and identify potential biases or limitations. |
| Efficiency | The model should be able to train and make predictions efficiently, especially for large or complex datasets. |
| Relevance | The model should be relevant to the task at hand and should address the specific problem or question being posed. |
| Novelty | The model should provide new insights or solutions that were not previously known or available. |
| Impact | The model should have a positive impact on the real world, either by solving a problem, improving a process, or making a decision. |
| Scalability | The model's ability to maintain performance as the size of the dataset increases. |
| Fairness | Ensuring the model does not create or reinforce unfair bias against certain groups. |
| Transferability | The ability of the model to adapt to different tasks or domains with minimal adjustments. |
| Compliance | Adhering to legal and ethical standards, especially in sensitive areas like healthcare or finance. |
| Sustainability | Evaluating the environmental impact of training and deploying the model, such as energy consumption. |
| Security | Ensuring the model is resistant to attacks and does not expose sensitive data. |
| User Experience | The ease with which end-users can interact with the model and its outputs. |
| Reproducibility | The ability for other researchers or practitioners to recreate the model and achieve similar results. |

Artificial neural networks (ANNs) have been used massively in the last few decades for a wide range of applications, particularly in the fields of computer vision, natural language processing, and robotics (see [6–8])The original ANN's architecture—at least in the initial proposal [9]—is loosely based on the way biological neurons work. Following some initial difficulties in simulating the boolean XOR operator [10], ANNs (see, for example, [11] for the historical path) have evolved to more advanced formalisms, such as convolutional neural networks (CNN). The latter is usually employed in the field of computer vision, for example, to recognize hand-written symbols. More advanced connective models refer to long-short term memory (LSTM), which are empowered with the ability to remember

the sequence of their inputs. As a result, they proved to be successful in recognizing human written text [12] by preserving the context of a sentence or even forecasting time series and predicting data patterns (see [13]).

This article presents the following claims:

- The argument regarding the innovative character of ML techniques should be taken with caution, as some of the deployed techniques are rather old and usually originate from the field of statistics;
- The argument concerning ML regarded as science does not hold entirely, as some of the most important ML models (i.e., ANNs) lack explanatory capabilities, although they preserve powerful prediction powers;
- The term "machine learning" oversells the concept. Although ML models are trained to accomplish specific tasks, there is no evidence that the model has any kind of comprehension of the phenomenon under study. This contradicts the essence of learning itself, as human beings usually learn to perform a task by developing a partial or even complete understanding of the skill at hand;
- It is not universally true that the knowledge exhibited by an ML model is effective, as the data used to train the model might be unbalanced, biased, or insufficient to allow the model to generalize what it has learned;
- On the other hand, innovative ML techniques, such as reinforcement learning (RL) or even imitation learning (IL) have a resemblance with learning methods exploited by human beings studied in cognitive sciences.

The target of this work consists of ML researchers and computer engineers. However, the core of the discussion holds special significance for philosophers of science and researchers in epistemology. Specifically, this article aims to contribute to the ongoing dialogue about the epistemological foundations and the scientific status of ML, critically reviewing its core foundations, limitations, and current perspectives.

The article is organized as follows. Section 2 formalizes the most important ML connective models. Section 3 reviews the epistemological status of ML, and provides an analysis of its shortcomings. Section 4 reviews the related work. Finally, Section 5 concludes this work and suggests some lines of future work. Figure 1 offers a visual representation of the work's structure.

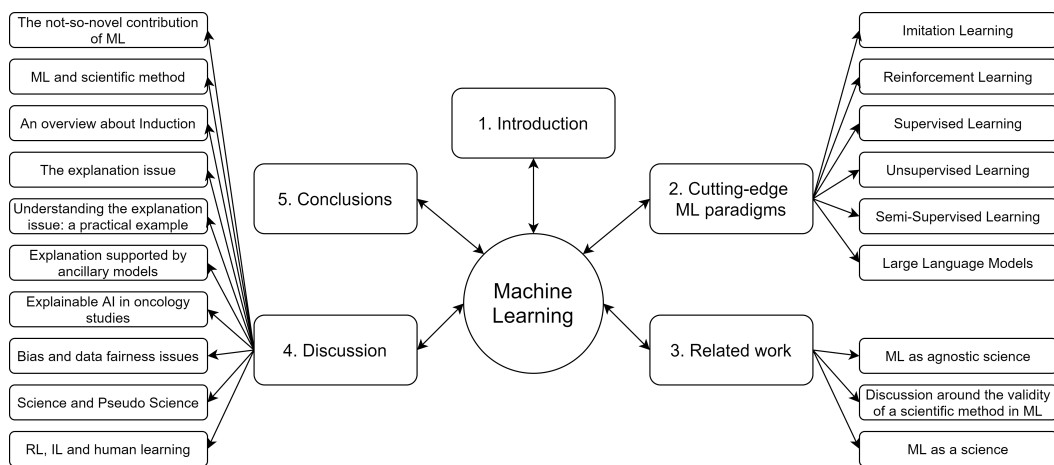

**Figure 1.** Visual representation of the work's structure.

## 2. Machine Learning Paradigms

This section formalizes the most important paradigms related to learning process in AI.

The continuous evolution of the ML field has yielded the introduction of additional standalone paradigms and methodologies. Within the spectrum of ML, imitation learning (IL), reinforcement learning (RL), supervised learning (SL), unsupervised learning (UL)

and semi-supervised learning (SSL) stand out as pivotal domains of investigation. They share the characteristic of enabling the machine to learn through a model, but they differ in how the learning operation is performed. Beyond these fundamental delineations, the AI research landscape encompasses a rich tapestry of methodologies such as self-supervised learning, active learning, transfer learning, distance learning, ensemble learning, Bayesian learning, structured learning, hierarchical learning, feature learning, metric learning, and continual learning. Each of these paradigms makes distinct contributions to the overarching goal of enabling computational systems to extract insights, make decisions, and adapt to diverse scenarios. Figure 2 shows the relationship between AI, ML, and the main advanced ML paradigms that are analyzed in this paper.

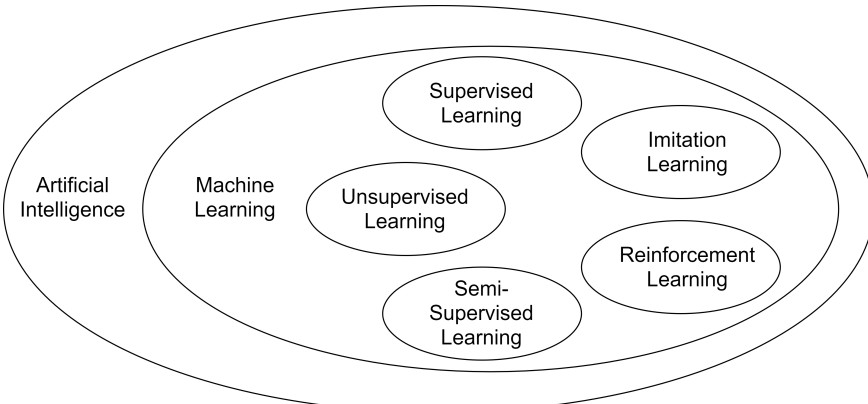

**Figure 2.** The relationship between AI, ML, and the main ML advanced ML paradigms.

### 2.1. Imitation Learning

Imitation learning (IL) comprises a learner that outputs responses by reproducing the observed behavior in the environment. Zheng et al. [14] propose a taxonomy of applications and challenges. The standard IL model includes the interaction between the following core elements: an agent, the environment, a teacher, and a policy. The agent (i.e., the learner) acquires information for its training. It deploys two main methods for learning a policy about its future actions: (i) learning from a teacher through a demonstration or (ii) learning from experience (using a reward or penalty factor to evaluate its progress). Regardless of the learning method, the agent must observe both the demonstration and the environment to acquire knowledge for its subsequent decision-making process. The IL learner is often composed of multiple spatial components, and therefore, its data include the coordinates of both the components and their joints, also potentially incorporating spatial limits. The environment comprises a finite and discrete set of stochastic states, $S = \{s_1, s_2, \ldots, s_n\}$, where each state represents the situation of the agent (position in the environment, information of its joints and status of a target). The agent is characterized by a discrete set of available actions $A = \{a_1, a_2, \ldots, a_n\}$, and interacts within the environment to achieve a given goal. The objective of the learning process is the development of a correct policy. The latter is denoted by a mapping between state $s$ and action $a$ to achieve a goal (i.e., desired behavior, or correct output). The agent uses the policy to decide which action $a_i$ it must take after the actual state $s_i$ of the environment is inputted. Learning through teacher demonstration works with a pair $(s_i, da_i)$, where the state $s_i$ is a vector of features in that instant, and the action $da_i$ is the performed action by the demonstrator (i.e., teacher). Learning through experience works instead with a reward (or a penalty, if negative) after an action $a_i$ is taken. It works with a tuple $(s_i, a_i, r_i, s_{i+1})$ of input state $s_i$, performed action $a_i$, awarded reward $r_i$ for the evaluation of the decision and new state $s_{i+1}$ of the environment. The main difference between the two types of learning is the inclusion of a teacher or a reward in the model. In both cases, the learning process involves capturing patterns and relationships within the data, allowing the agent to make informed decisions in similar future scenarios. The incorporation of a teacher provides explicit guidance, while reward-

based learning relies on consequential feedback from the environment, emphasizing the adaptability and autonomy of the agent in refining its decision-making capabilities. The outputted action is a vector of the parts of the agent that must be moved to change the environment state according to the learned decision-making policy. Figure 3 shows a schema of the setting of the IL model. The learner (i.e., the agent) is surrounded by the environment and uses the provided information from the demonstrator (i.e., the teacher) and the state $s_i$ from the environment as the input, to output an action $a_i$. The process is then repeated. Takayuki et al. [15] observe that the teacher is often human, making IL the best choice for transferring knowledge from humans to robots. IL performs optimally in autonomous control systems, and high-dimensional problems are solved efficiently. As outlined by Hussein et al. [16], one prominent application of IL involves the training of robots performing various actions, such as driving vehicles. In this context, a robot consists of multiple movable joints, and the state vector $s_i$ encapsulates their coordinates before any action is executed. The vector representing the subsequent state change, $s_{i+1}$, encompasses the new coordinates of these joints following the performed actions. Moreover, the authors underscore that, despite IL exhibiting superior performance in scenarios where other ML paradigms encounter challenges, it is not without limitations. For example, (i) high-performance results are limited to task-specific environments; (ii) effectiveness is highly dependent on the quality of the provided demonstrators; (iii) locally optimal solutions are more often reached than globally optimal solutions; (iv) policy representation issues impose penalties on both data and computational efficiency.

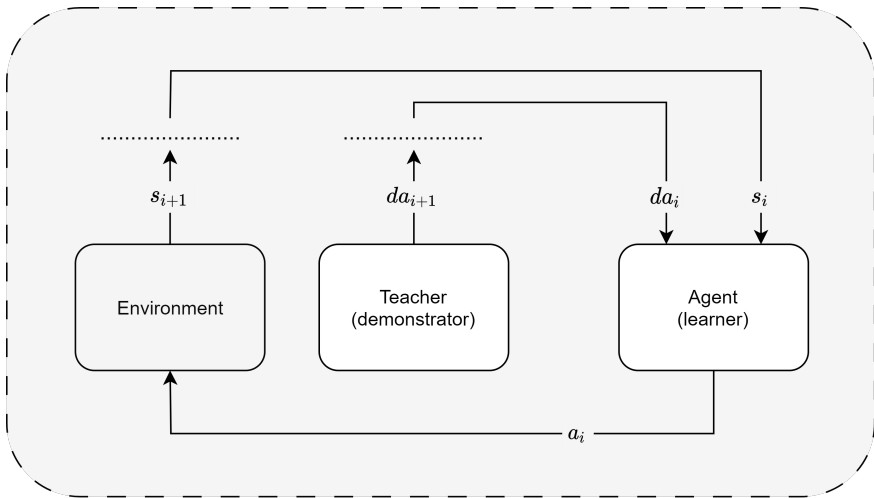

**Figure 3.** Imitation learning framework: a visual representation of the IL process.

### 2.2. Reinforcement Learning

Reinforcement learning (RL) comprises a learner who has to discover which actions (or subsequent groups of actions) to perform to reach the objective of maximizing an assigned numerical reward. RL is characterized by trial-and-error search and delayed reward phases. The standard RL model comprises the interaction between an agent, the environment, a policy, a reward, and a value function. The environment includes a finite and discrete set of stochastic states, $S = \{s_1, \ldots, s_n\}$. Furthermore, the agent is denoted by a discrete set of available actions, $A = \{a_1, \ldots, a_n\}$. The agent is the learner, as it pursues a given goal through its decision on which action to perform at each step. Agent and environment are connected, as the agent is situated within the environment and interacts with it (by changing its state). The agent receives the current state $s_i$ of the environment as an input. Depending on the action that can return the highest reward $r_i$, the agent behaves accordingly and produces an output (i.e., the action $a_i$). The future state of the environment depends on the given current state and on the chosen action. The reward $r_i$ serves as the signal from the environment to the agent after each interaction, indicating the goodness of the action taken in a specific state. In reinforcement learning, Kaelbling et al. [17] analyzed algorithms

for systematic trial-and-error searches to discover successful combinations of actions that maximize the cumulative reward (i.e., maximize the expected measure of reinforcement). The value function, denoted as $V(s)$ represents the expected cumulative reward from a given state $s$ onwards. The agent aims to maximize this expected cumulative reward by selecting actions that lead to higher values of the value function. The goodness of the model's performance can be evaluated based on the final values of the value function across different states. Figure 4 shows a schema of the setting of the RL model. The learner (i.e., the agent) is surrounded by the environment and uses its state $s_i$ and its reward $r_i$ as the input to output an action $a_i$. The process is then iterated. Despite RL's crucial role in various applications, it has some major drawbacks that negatively impact its performance. Casper et al. [18] investigate the issue of inputted human feedback, stating that there exist challenges with the human feedback, with the reward model, and with the policy. Humans cannot evaluate difficult tasks well and can be easily misled. The reward function cannot represent different humans from different societies uniquely without introducing bias. Last, a correctly developed policy can pursue the wrong goal when the true goal is in reality correlated with other factors. Li [19] presents an extensive taxonomy outlining the diverse applications of RL in real-world scenarios. In particular, Dworschak et al. [20] investigate the relationship between RL and design automation, concluding that RL's feasibility, training effort, and transferability show encouraging results.

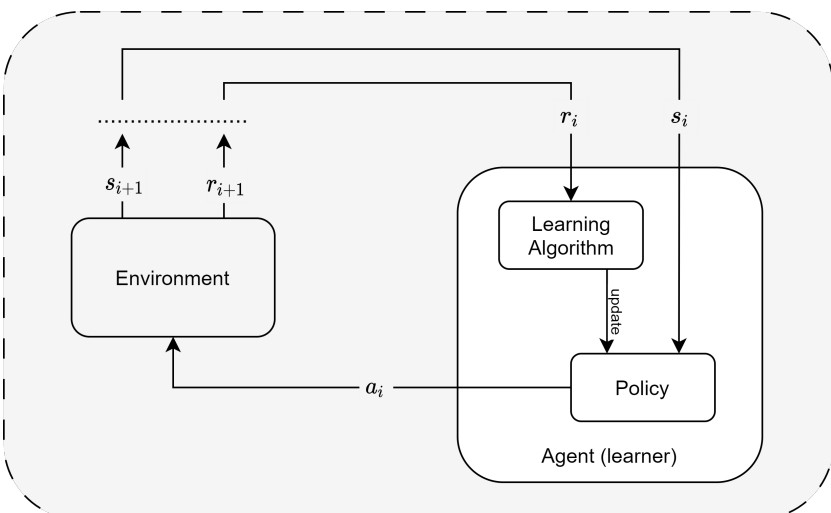

**Figure 4.** Reinforcement learning framework: a visual representation of the RL process.

### 2.3. Supervised Learning

Supervised learning (SL) is an ML paradigm where the algorithm is inputted with a labeled dataset. A defining feature of this method lies in its provision of both observations and corresponding desired outputs to the learner. The model is trained on input–output pairs, denoted as $(x_i, y_i)$, where $x_i$ is the input and $y_i$ is the corresponding desired output. The primary goal of supervised learning is to establish a mapping between input and output, enabling the accurate prediction of unseen data in the future. The algorithm evaluates its predictive accuracy by employing a loss function, quantifying the disparity between predicted outcomes and actual labels. The loss function $L(\hat{y}, y)$ represents the extent of the model's deviation from the true values, where $\hat{y}$ signifies the predicted output and $y$ denotes the actual label. The learning process involves the algorithm adjusting its parameters iteratively to minimize the discrepancy between its predictions and the true labels. When the error, quantified by the loss function, reaches an acceptable threshold, the model is deemed sufficiently trained and therefore able to generalize and accurately predict outcomes for new, unseen data instances. Figure 5 shows a schema of the setting of the SL model. The learner obtains the data $x_i$ as the input and produces an output $\hat{y}$. The output is compared with the true value $y_i$ via the calculation of the loss function

$L(\hat{y}_i, y_i)$. The process is then iterated. SL performs excels in classification tasks, as Ayodele [21] shows. SL is particularly suitable for the task of training ANNs, as they are highly dependent on the retrieved knowledge from the inputted data (i.e., on the available true values of outputs). Mehlig [22] explores classification tasks solved through ANNs trained using labeled datasets inputted to SL. The perceptron learning process involves the adjustment of the model weights and biases based on the loss function. The output of a perceptron where observations and weights are inputted is produced through an activation function. As a result, the learning process involves adjusting the weights and biases of perceptrons to minimize the loss function value. As highlighted by Liu et al. [23], SL's success extends across diverse domains, including information retrieval, data mining, computer vision, speech recognition, bioinformatics, cheminformatics, and market analysis. However, the authors caution that achieving low training error does not necessarily guarantee optimal performance during testing due to the imperative need for generalizability, aiming to avoid both underfitting and overfitting. One of the primary limitations of SL lies in its requirement for labeled data during the input phase, necessitating pairs of inputs and corresponding outputs for model training. To mitigate this constraint, semi-supervised learning (SSL) and unsupervised learning (UL) present alternatives that demand, respectively, fewer or no output labels as the inputs for training.

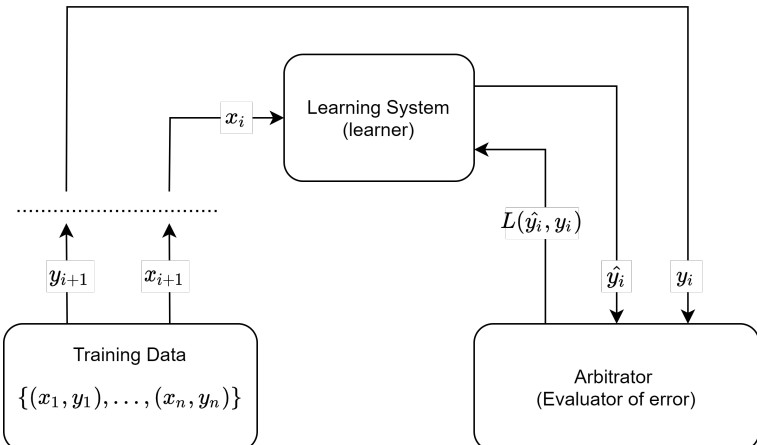

**Figure 5.** Supervised learning framework: a visual representation of the SL process.

### 2.4. Unsupervised Learning

Unsupervised learning (UL) assumes that learning occurs through inputted unlabelled data into an algorithm. The unsupervised paradigm works by finding patterns, relationships, grouping, and structures within the inputted data without any guidance. Therefore, the input data are $X = \{x_1, x_2, \ldots, x_n\}$ and the algorithm must autonomously identify meaningful representations (i.e., hidden patterns or relationships) within the data. Naeem et al. [24] explain that UL makes machines more intelligent as it enables them to learn independently using models that detect trends and patterns and make decisions based on them. Notable UL applications concern clustering and dimensionality reduction. Clustering a dataset means partitioning the data into clusters (i.e., groups) such that similar instances $x_i$ are grouped. In a clustering model, $C = \{c_1, c_2, \ldots, c_k\}$ represent the set of clusters, where each $c_i$ is a subset of $X$. The algorithm aims to minimize an objective function, often involving the within-cluster variance or a measure of similarity. Mathematically, it is expressed as

$$\min_{C} \sum_{i=1}^{k} \sum_{x \in c_i} D(x, \mu_i) \tag{1}$$

where $\mu_i$ is the centroid or representative of cluster $c_i$, and $D()$ is a distance metric. In dimensionality reduction, the goal is to represent the data in a lower-dimensional space while preserving its essential characteristics. The objective is to find a mapping function

that minimizes the reconstruction error (i.e., the dissimilarity between the original data points and their new versions).

$$\min_{f} \sum_{i=1}^{n} \|x_i - f(x_i)\|^2 \tag{2}$$

Figure 6 shows a schema of the setting of the UL model. The learner obtains the data $x_i$ as the input and produces an output $C$ (for example, the clusters), satisfying the designed evaluation metric. One notable advantage of employing UL algorithms lies in their inherent suitability for handling unstructured data. Many UL algorithms are specifically designed to navigate and extract meaningful patterns from datasets that lack predefined labels or organized structures. This adaptability makes UL particularly effective in scenarios where the data are unorganized, allowing these algorithms to discern underlying structures, relationships, or clusters without the need for explicit supervision.

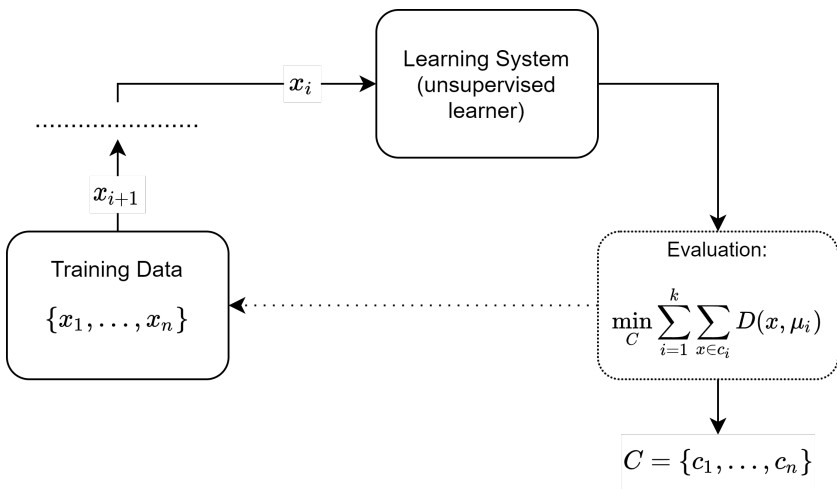

**Figure 6.** Unsupervised learning framework: a visual representation of the UL process.

### 2.5. SemiSupervised Learning

Semi-supervised learning (SSL) is a hybrid form that combines both UL and SL, as the inputted data are a mixture of both input-only and input–output couples. SSL is advised in situations where SL is required, but obtaining a complete dataset is not possible (for example, for expensive issues). Again, the objective is to learn a pattern to classify future available data. In addition to unlabelled data $X = \{x_1, x_2, \dots, x_n\}$, the algorithm is provided with some supervision information (i.e., labelled data) $X' = \{(x_1, y_1), (x_2, y_2), \dots, (x_n, y_n)\}$. Chapelle [25] explains the importance of SSL: it can be used for a variety of tasks where most data are unlabelled. A disadvantage in using SSL is that, as unlabelled data carry less information than labelled data, to increase prediction accuracy significantly, a great amount of input data is required. An advantage is instead that the presence of unlabelled data helps in improving the model's generalization and performance. Figure 7 shows a schema of the setting of the SSL model. The learner obtains the data $x_i$ as the input and produces an output $\hat{y}_i$. The output is confronted with the true value $y_i$ (if available) via the calculation of the loss function $L(\hat{y}_i, y_i)$. The process is then repeated. Note that the line of the ground truth is dotted, as most observations are not inputted with their expected output.

### 2.6. Comparison of the Proposed ML Paradigms

The analyzed ML paradigms share some common features although they present also major differences.

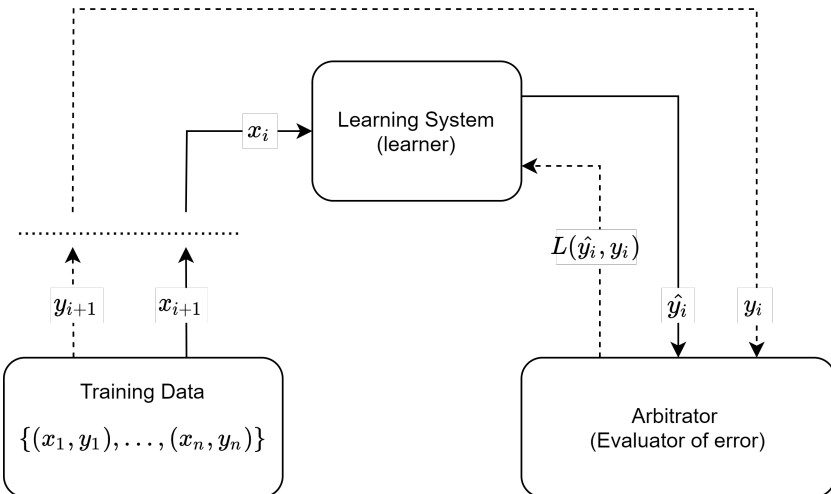

**Figure 7.** Semi-supervised learning framework: a visual representation of the SSL process.

In RL, the focus is on training an agent to make decisions within a given environment. The agent learns by performing actions and receiving rewards or penalties, thereby developing a strategy or policy to maximize cumulative rewards over time. This method is particularly effective in scenarios that require a balance between exploration and exploitation, such as in-game playing (such as chess or go), autonomous vehicle navigation, robotics, and recommendation systems. RL does not require labeled data; instead, it relies on reward feedback from the environment. SL, on the other hand, is centered around learning a function that maps inputs to outputs. This approach is suitable for tasks such as image and speech recognition, weather forecasting, and medical diagnosis, where the goal is to predict the output for a new input based on the learned function. The effectiveness of SL largely depends on the quantity and quality of the labeled data available for training. UL diverges from these paradigms by focusing on identifying patterns or structures in unlabeled data. Without explicit outcome labels to guide the learning process, UL algorithms seek to organize the data in some meaningful way. This form of learning is useful in a variety of applications, including market basket analysis, gene sequencing, social network analysis, and organizing extensive document libraries, where the underlying patterns within the data need to be identified. SSL strikes a balance between supervised and unsupervised learning. It is particularly useful when there is a limited amount of labeled data augmented by a larger quantity of unlabeled data. By combining elements from both SL and UL, SSL can improve learning accuracy. This approach is often implemented in scenarios like web content classification, language translation, and speech analysis, where acquiring a large set of labeled data can be challenging or very expensive.

Table 2 reports a comparison between RL, UL, IL and SL. The table does not explicitly show data about SSL, as it is a hybrid form of UL and SL.

### 2.7. Large Language Models

The emergence and evolution of large language models (LLMs) like GPT-3 in advanced machine learning represent a paradigm shift in natural language understanding and generation. These models, trained on extensive corpora of text data, demonstrate an unprecedented ability to process and produce human language. This capability has far-reaching implications across various domains [26,27]. In terms of language understanding, LLMs have set new benchmarks in tasks like sentiment analysis, topic classification, and contextual interpretation. Their Deep Learning (DL) architectures enable them to grasp nuanced language features, making them adept at understanding context, irony, and even cultural references in text. This has significant applications in areas like social media monitoring, market analysis, and cultural studies, where understanding the subtleties of human communication is crucial. Regarding language generation, LLMs have shown proficiency

in creating coherent and contextually relevant text, ranging from writing assistance to generating creative content. They are being used in novel applications like generating news articles, writing poetry, and even scripting for virtual assistants. This capability has opened new avenues in content creation, where the ability to generate diverse and sophisticated text is invaluable. Moreover, the adaptability of LLMs in handling various language styles and formats is notable. From formal reports to casual conversations, these models can tailor their outputs to fit different linguistic styles, making them versatile tools in fields such as customer service, where they can interact with users naturally and engagingly. In the educational domain, LLMs are transforming the landscape by offering personalized learning experiences. They can provide explanations, solve problems, and interact in an educational dialogue, making them valuable assets in digital learning platforms. However, with these advancements come challenges and responsibilities. Issues such as data bias, ethical considerations in automated content generation, and the potential impact on jobs in content-related fields are areas of ongoing research and debate. Ensuring responsible use and continuous improvement of these models is paramount to leverage their potential positively. LLMs represent a significant milestone in machine learning, pushing the boundaries of what machines can understand and express in human language. As they continue to evolve, their impact is expected to grow, reshaping the interaction between humans and technology.

**Table 2.** Comparison of RL, UL, IL and SL.

| Aspect | RL | UL | IL | SL |
|---|---|---|---|---|
| Learning Objective | Optimal Policy | Discover Hidden Structures | Imitate Expert Behavior | Mapping Input to Output |
| Input Data | State and Reward Signals | Unlabelled Raw Data | Demonstrations, State-Action Pairs | Labelled Input-Output Pairs |
| Interaction with Environment | Sequential Decision Making | No Explicit Interaction | Observing Demonstrations | No Interaction |
| Feedback Signal | Reward Signals | Evaluation Metrics or Criteria | Expert Demonstrations, Rewards | Correct Output Labels |
| Key Mechanism | Exploration-Exploitation and Policy Learning | Clustering, Dimensionality Reduction | Imitation and Policy Learning | Pattern Recognition |
| Training Approach | Trial and Error | Self-Organization | Learning from Expert Behavior | Error Minimization |
| Applicability | Sequential Decision Making | Pattern Discovery | Skill Transfer and Imitation | Classification, Regression |
| Examples of Applications | Game Playing, Robotics | Clustering, Dimensionality Reduction | Autonomous Vehicles, Robotics | Image Recognition, Speech Recognition |

### 3. Discussion

Although the issues of bias, fairness, and explainable AI are well-known in the ML community, their implications and challenges evolve with advancements in technology and applications. This section revisits these issues in light of recent developments, new research findings, and emerging technologies, showing how these perennial concerns are manifesting. Furthermore, it is necessary to stress the urgency of continuous discussion about these issues in the ML community. The fact that these issues are well-known does not diminish their importance; ongoing dialogue is crucial for developing better practices, updating policies, and educating newcomers to the field. In this sense, this section makes these issues accessible or relevant to a broader, perhaps non-specialist audience.

The implications of bias, fairness, and explainability in ML extend beyond the technical community to sectors like policy-making, legal affairs, and public perception. Furthermore, it is important to clarify the epistemological position of ML, what is novel, and what is derived from other disciplines.

### 3.1. The Not-so-Novel Contribution of ML

ML mostly employs methods from statistics or even from other scientific areas (for example, data clustering was proposed initially in anthropology by Boas [28] and then Murdock [29] to identify cultural patterns and understand the relationships between different cultural groups). The recent availability of vast amounts of data joined with more powerful and affordable hardware, allowed researchers to exploit consolidated statistical methods from a new perspective.

Following this direction, ML embodies a synthesis of historical theoretical foundations and contemporary advancements. While the field relies on well-established statistical principles dating back to the late 18th century, its contemporary significance is derived from the amalgamation of these classical theories with novel methodologies and unprecedented computational capabilities.

The foundational algorithms, such as linear regression and K-nearest neighbors, underscore the enduring relevance of classical statistical concepts.

As a result, the novelty of ML consists mostly in the adoption of advanced ANNS formalisms (see Table 3).

**Table 3.** ML models in chronological order.

| Model | Year | Description |
| --- | --- | --- |
| Linear Regression | Late 18th century (method of least squares) | The principles of linear regression, a fundamental statistical technique, were established in the 18th century. |
| Principal Component Analysis (PCA) | 1933 (formalized) | The concept of principal component analysis dates back to the early 20th century, and it was formally introduced by Karl Pearson in 1933. |
| Neural Networks (Perceptron) | 1940s | Neural network concepts trace back to the 1940s with the introduction of the perceptron. The resurgence of neural networks with DL occurred around the 2010s, facilitated by advances in computing power and data availability. |
| Naive Bayes | 1950s | The foundations of the Naive Bayes classifier were laid in the 1950s, and it has since been widely used in classification tasks. |
| K-Nearest Neighbors (K-NN) | 1950s | The K-NN algorithm, while not formalized as it is today, has its roots in the 1950s. It became more widely used in pattern recognition and classification in subsequent decades. |
| K-means Clustering | 1950s | K-means is a foundational clustering algorithm developed in the 1950s. It partitions data into k distinct clusters based on the mean distance to the centroids. |
| Hierarchical Clustering | 1950s | Hierarchical clustering, developed in the 1950s, creates a tree of clusters by either merging smaller clusters into larger ones or splitting larger clusters. |
| k-Medians Clustering | 1950 | An adaptation of k-means, k-medians clustering uses medians instead of means, which can provide robustness to outliers in the dataset. |
| Perceptron | 1957 | The first and simplest type of artificial neural network devised by Frank Rosenblatt. |
| Hidden Markov Models (HMM) | 1960s | Hidden Markov Models were introduced in the 1960s by Leonard E. Baum and others, primarily for applications in speech recognition. |

Table 3. *Cont.*

| Model | Year | Description |
|---|---|---|
| Decision Trees | 1960s | The concept of decision trees was introduced in the 1960s, with the development of algorithms like ID3 (Iterative Dichotomiser 3) by Ross Quinlan. |
| Backpropagation Network | 1970s | Popularized in the 1980s, this network employs the backpropagation algorithm for training, crucial for DL. |
| Mean Shift | 1975 | Developed in 1975, Mean Shift is a non-parametric clustering technique used for locating the maxima of a density function. |
| Expectation Maximization (EM) | 1977 | Introduced in 1977, EM is an iterative method used for finding maximum likelihood estimates in statistical models, particularly in the presence of latent variables. |
| Convolutional Neural Network (CNN) | 1980s | Pioneered by Yann LeCun, CNNs excel in processing data with a grid-like topology, such as images. |
| Recurrent Neural Network (RNN) | 1980s | Designed to recognize patterns in sequences of data, such as text or time series. |
| Support Vector Machines (SVM) | 1990s | The formulation of support vector machines for classification and regression tasks was introduced by Vladimir Vapnik and his colleagues in the 1990s. |
| DBSCAN | 1996 | DBSCAN, introduced in 1996, is a density-based clustering algorithm that groups points that are closely packed together, marking outliers in less dense regions. |
| Long Short-Term Memory (LSTM) | 1997 | An advanced RNN variant, capable of learning long-term dependencies, introduced by Sepp Hochreiter and Jürgen Schmidhuber. |
| Spectral Clustering | 2000s | Spectral clustering, gaining popularity in the 2000s, uses eigenvalues of a similarity matrix to reduce dimensionality before clustering in fewer dimensions. |
| Random Forest | 2001 | The random forest algorithm was proposed by Leo Breiman in 2001, extending the concept of decision trees. |
| Gradient Boosting Machines | 2001 | The concept of gradient boosting, the foundation for algorithms like AdaBoost and XGBoost, was proposed by Jerome Friedman in 2001. |
| Generative Adversarial Network (GAN) | 2014 | Proposed by Ian Goodfellow, GANs are used for generating data, particularly in image generation tasks. |
| Transformer Networks | 2017 | Introduced in the paper "Attention Is All You Need", transformers revolutionized natural language processing. |

The prediction power is one of the key aspects of ML methods. Improving the prediction accuracy of ML models is a dynamic field where several intertwined strategies are employed. At the core of enhancing ML models is the focus on data quality and quantity. High-quality, relevant, and comprehensive datasets form the bedrock upon which effective models are built. To complement this, feature engineering and selection play a crucial role, as identifying the most impactful features can significantly influence a model's predictive power. The complexity and architecture of the model itself are also crucial. While more complex models, such as deeper neural networks, can capture intricate patterns in data, it is a delicate balance to maintain to avoid overfitting. In this context, ensemble methods, which combine predictions from multiple models, emerge as a powerful approach to achieving more accurate and robust predictions. Hyperparameter optimization is another key area of focus. By fine-tuning the model's hyperparameters through techniques like grid search, random search, and Bayesian optimization, significant performance improvements can be realized. This is complemented by advanced algorithms, particularly in DL, that have

set new standards in various domains like image processing, sequential data analysis, and natural language processing. Transfer learning, where a model trained on one task is adapted for another, is especially effective in scenarios where labeled data are limited. Regularization techniques such as dropout and L1/L2 regularization are also employed to prevent overfitting, enhancing the model's generalization capabilities on new data. The concept of real-time learning and adaptation, where models continually update and adapt to new data, is gaining traction for its ability to improve predictive accuracy over time. Finally, adopting cross-disciplinary approaches that incorporate insights and techniques from fields like statistics, cognitive science, and physics is leading to the development of innovative and more accurate ML models. See, for example, Zewe (https://news.mit.edu/ 2023/improving-machine-learning-models-reliability-0213, visited on 5 January 2024), who developed a technique for effective uncertainty quantification in machine learning models. This method, which does not require model retraining or additional data, uses a simpler companion model (metamodel) to assist the original model in estimating uncertainty. This approach focuses on both data and model uncertainty, helping in better decision making and trust in model predictions.

### 3.2. ML and Scientific Method

The scientific method provides a systematic and rigorous framework for investigating the natural world. It is a cyclical process that encompasses observation, hypothesis formulation, experimentation, data analysis, and conclusions. The formulated hypotheses are tentative explanations that attempt to account for the observed phenomena. As a result, hypotheses are informed by existing knowledge, scientific principles, and logical reasoning. Well-defined hypotheses are specific, testable, and falsifiable, meaning they can be proven wrong through experimentation. Experimentation is the crucible of the scientific method. In particular, it is the process of testing hypotheses against empirical evidence. Experiments are carefully controlled, isolating variables and minimizing biases to ensure the reliability of the results. The raw data collected from experiments undergo meticulous analysis and interpretation. Scientists employ statistical techniques, mathematical models, and visualization tools to extract meaning from the data, identifying trends, patterns, and correlations. Data analysis provides crucial insights into the validity of the hypotheses. Based on the analysis of experimental data, it is possible to conclude the validity of the hypotheses. If the data support the hypotheses, the conclusions provide provisional explanations for the observed phenomena. However, if the data contradict the hypotheses, the conclusions lead to their rejection or modification, prompting further investigation. The scientific method is not a linear process; it is an iterative cycle of observation, hypothesis formulation, experimentation, data analysis, and conclusion formation. As new evidence emerges, existing theories are refined, and new hypotheses are formulated, leading to a continuous advancement of scientific knowledge. The essence of the scientific method lies in its objectivity, rigor, and self-correcting nature. It is a process that embraces doubt, skepticism, and critical thinking, ensuring that scientific knowledge is not based on dogma or personal beliefs but on empirical evidence and logical reasoning.

It can be argued that ML can be considered an extension of the scientific method, particularly in its use of data-driven hypothesis testing and iterative refinement. ML algorithms are trained on vast amounts of data, from which they extract patterns and make predictions. This process resembles the scientific method's emphasis on observation, hypothesis formulation, and experimentation. Additionally, ML models are continuously evaluated and refined based on their performance on new data, mirroring the scientific method's iterative nature. As new data become available, ML models can be updated to improve their accuracy and generalizability. AI and ML are being increasingly integrated into scientific discovery [30], helping scientists generate hypotheses, design experiments, and collect data. It highlights instances where AI has significantly advanced fields like pure mathematics and molecular biology, demonstrating the predictive power and the ability to guide human intuition in complex scientific challenges. Boge et al. [31] review the role

of ML as an optimization tool executed by digital computers and how its increasing use signifies a shift from traditional scientific aims of explanation towards pattern recognition and prediction. It explores the implications of this shift for scientific explanation and the potential future directions of ML in scientific research. Finally, Buchholz et al. [32] discuss some of the insights into the use of ML models as a method for science outside of traditional success areas. It opens up a debate on how ML coordinates with other scientific methods, transitioning from explanatory to predictive models, and how it applies in various scientific domains like AlphaFold in protein folding prediction.

Opponents of this view argue that ML represents a distinct approach to knowledge discovery, departing from the traditional scientific method in several ways. While ML excels at pattern recognition and prediction, it cannot often provide causal explanations for the observed patterns. The "black box" nature of many ML models makes it challenging to understand the underlying mechanisms driving their predictions. Furthermore, ML algorithms are often trained on data that may contain biases or inaccuracies, which can lead to biased or erroneous predictions. These biases can be difficult to detect and eliminate, particularly in complex models. Table 4 summarizes some of the most important similarities between the scientific method and ML.

**Table 4.** A comparison between scientific method and ML aspects.

| Aspect | Scientific Method | ML |
| --- | --- | --- |
| Goal | Understand the natural world and uncover its underlying principles | Make predictions or classifications based on patterns observed in data |
| Approach | Data-driven, but with a strong emphasis on theoretical understanding and causal explanations | Data-driven, with a focus on empirical patterns and correlations |
| Methods | Observation, experimentation, hypothesis testing, mathematical modeling, and rigorous analysis | Statistical analysis, data mining, ML algorithms, and pattern recognition |
| Evaluation | Predictive accuracy, replicability, falsifiability, and explanatory power | Predictive accuracy, generalizability, interpretability, and robustness |
| Explanation | Strives to provide causal explanations for observed phenomena | May not always be able to provide causal explanations, but can provide insights into correlations and patterns |
| Iteration | Iterative process of observation, hypothesis testing, and refinement, driven by theoretical understanding and empirical evidence | Iterative process of training, evaluation, and refinement, driven by data availability and performance optimization |
| Limitations | Limited by the scope of human understanding and the ability to design and execute meaningful experiments | Limited by the quality and quantity of data, the inherent biases in data, and the complexity of real-world problems |
| Strengths | Provides a robust framework for uncovering causal relationships and understanding the natural world | Enables efficient and accurate predictions in complex domains, including applications in healthcare, finance, and technology |
| Relevance | Essential for understanding the fundamental principles that govern the universe and making informed decisions about the world around us | Plays a crucial role in solving real-world problems and driving innovation in various fields |

Whether a branch of science that cannot causally explain a fact but can predict facts can still be considered science depends on the specific definition of science being used. A narrow definition of science might require that all scientific explanations be causal, meaning that they must explain the underlying mechanisms that cause the observed phenomena. Under this definition, a branch of science that cannot provide causal explanations would not be considered true science. However, a broader definition of science might allow for non-causal explanations, as long as they are still based on empirical evidence and rigorous methodology. Under this definition, a branch of science that can predict facts, even if it cannot explain them causally, could still be considered science. For example, meteorology can predict the weather with a high degree of accuracy, but it cannot fully explain the

complex mechanisms that drive weather patterns. However, meteorology is still considered a science because its predictions are based on empirical evidence and rigorous methodology.

### 3.3. An Overview about Induction

Within epistemology, induction stands as a mode of reasoning that delves into the realm of generalities, extracting overarching principles or patterns from a constellation of specific observations or instances. This inductive approach navigates from the particular to the universal, seeking to discern broader truths from empirical evidence. However, as observed by the philosopher David Hume, the very foundation of induction harbors inherent logical complexities. Consider the plight of an individual who has only encountered white swans throughout their existence. Induction would compel them to posit a universal conclusion: all swans are white. This inference, distilled from repeated encounters with white swans, is then extrapolated to encompass the entirety of swans, both observed and unobserved. However, the validity of this conclusion rests upon the premise that future instances will mirror past observations. Philosophically, this challenge manifests as the "problem of induction", as articulated by Hume. The mere observation of white swans does not conclusively guarantee that all swans, including those yet to grace our presence, will share the same hue. The inductive process hinges upon the assumption that the future mirrors the past, a notion that introduces an element of uncertainty.

Mathematical induction functions as a foundational element within the realm of mathematical proofs, facilitating the meticulous establishment of universally applicable assertions through finite observations. The crux of mathematical induction is encapsulated in its dual-phased structure, consisting of a base case and the inductive step. The former is a fundamental stage that necessitates the verification of the proposition's validity for the smallest element within the designated set. For instance, to substantiate the claim that each even number can be represented as the sum of two consecutive integers, one must validate this assertion for the smallest even number, namely 2. The inductive step entails demonstrating that the proposition remains valid for any natural number after the base case. To extrapolate the aforementioned even number proposition, it is imperative to illustrate that if any even number, denoted as 'n', can be expressed as the sum of two consecutive integers, then its successor, represented as 'n + 2', can similarly be articulated in the same fashion. Philosophical scrutiny of mathematical induction unveils its inherent strengths and limitations. On the one hand, it furnishes a rigorous framework for deriving universal truths from finite observations, a task that would prove insurmountable through deductive reasoning alone. The capacity to generalize from specific instances constitutes a pivotal aspect of mathematical exploration and discovery.

Within the domain of inductive reasoning, it is of paramount importance to recognize a fundamental epistemological principle: the conclusion derived from an inductive argument does not possess an inherent necessity for truth, even when the premises themselves are demonstrably accurate. Unlike deductive reasoning, where the truth of premises unequivocally guarantees the truth of the conclusion, inductive reasoning does not provide such absolute assurance. This epistemological uncertainty arises from the very nature of inductive inference, which hinges on the generalization from specific instances to formulate a broader statement. The extrapolation of a universal claim based on observed particulars inherently introduces an element of probabilistic reasoning rather than logical necessity. As such, the veracity of the conclusion remains contingent upon the assumption that future instances will align with past observations—an assumption that is inherently susceptible to doubt. Philosophically, this characteristic of inductive reasoning highlights the distinction between justification and truth. While inductive arguments can provide a plausible justification for a conclusion based on available evidence, the conclusion's truth cannot be definitively guaranteed, as it is predicated on the unverifiable assumption that the future will conform to past patterns.

Moreover, induction is not objective. The conclusions of inductive arguments are based on subjective judgments about the weight of evidence. Consider a scenario where

a person has sampled a few restaurants in a new city and, based on their subjective experiences, concludes that all the restaurants in that city must offer exceptional cuisine. The individual's judgment about the quality of the sampled restaurants forms the basis of an inductive inference that extends to all other restaurants in the city.

In this case, the conclusion drawn—namely, that all restaurants in the city provide exceptional cuisine—is heavily dependent on subjective judgments. The individual's assessment of "exceptional" is inherently subjective, influenced by personal tastes, preferences, and experiences. The weight assigned to the limited sample of restaurants may not accurately represent the diversity or variation in culinary quality across the entire city.

The subjectivity in evaluating the evidence becomes evident when considering that another person with different tastes or experiences might visit a different set of restaurants and reach a contrasting conclusion about the overall quality of the city's dining establishments. This example highlights how inductive reasoning, especially when based on subjective assessments, can lead to conclusions that lack objectivity and may not be universally applicable.

Induction serves as a foundational concept in the field of ML, particularly in its early stages, as it underpins the process of learning patterns and making predictions from data. ML algorithms rely on induction to generalize from observed examples and make predictions or decisions about unseen instances. This approach is particularly valuable when dealing with complex and unstructured data where explicit programming may be challenging. One notable example of an ML algorithm based on induction is the decision tree algorithm. Decision trees recursively partition the data based on features, creating a tree-like structure where each node represents a decision based on a specific feature. The algorithm learns these decision rules from labeled training data, making it capable of predicting the class or outcome of new, unseen instances. Another example is the K-nearest neighbors (K-NN) algorithm, which makes predictions based on the majority class of the K-nearest data points in the feature space. In this case, the algorithm induces patterns from the relationships between data points, allowing it to classify new instances based on their proximity to existing examples. Furthermore, support vector machines (SVMs) are a class of algorithms that learn decision boundaries by maximizing the margin between different classes in the feature space. SVMs, in their essence, rely on the inductive process of generalizing from training data to classify new data points.

In the context of strict ML, which primarily involves learning patterns from data to make predictions or decisions, the concept of induction remains a fundamental aspect. Inductive reasoning, or generalizing from specific examples to make predictions about new instances, is inherent in the learning process of ML models. ML models are designed to recognize patterns and relationships within data, and this recognition typically involves the inductive inference that observed patterns will generalize to unseen instances. The examples provided earlier, such as decision trees, K-nearest neighbors, and support vector machines, all rely on inductive reasoning to make predictions. It is important to note that the nature of ML is such that models learn from data, and this learning process inherently involves generalization. The very essence of ML is to capture underlying patterns in the data to make predictions on new, unseen instances. If an ML model did not employ some form of induction or generalization, it would struggle to make meaningful predictions on new data.

### 3.4. The Explanation Issue

Although some models such as polynomial regression and decision trees are considered "white box" because they are transparent and it is possible to determine the weight of predictors variable in predicting a target variable, some other formalisms, such as ANNs and random forests, are regarded as "black box". Both explanation and interpretation play a crucial role in regarding the problem of providing causal explanations in ML models. Explanation refers to the process of clarifying or providing a comprehensible account of how a particular decision or outcome was reached by an AI model. It involves breaking

down complex model predictions or decisions into understandable terms for users or stakeholders. The primary purpose of explanation in Explainable AI (XAI) is to enhance transparency, trust, and accountability. Providing clear explanations helps users, especially those without a deep understanding of ML, to grasp why a model made a specific prediction or decision. On the other hand, interpretation, in the context of XAI, involves understanding and making sense of the internal workings of an ML model. It delves into uncovering the features, variables, or patterns that contribute significantly to the model's predictions. Interpretation goes beyond the surface-level explanation by exploring the internal mechanisms of the AI model. It helps data scientists and researchers gain insights into how the model is processing information and which features are influential in its decision making. Both explanation and interpretation contribute to the overarching goal of XAI, which is to make AI systems more transparent, understandable, and accountable. While explanation targets end-users and stakeholders, interpretation is often a concern for data scientists and researchers seeking to improve the robustness and reliability of ML models. The benefits of XAI are numerous: (i) increased trust and transparency, as it can help to increase trust in AI systems by making them more transparent and understandable to humans; (ii) improved decision making, as it can help humans make better decisions by providing them with insights into how AI systems work; (iii) reduced bias, as XAI can help to reduce bias in AI systems by making it easier to identify and correct biases in the data and the algorithms; and finally, (iv) increased safety XAI can help to increase the safety of AI systems by making it easier to identify and mitigate risks.

*3.5. Understanding the Explanation Issue: A Practical Example*

The following example aims to clarify the fundamental issue of balancing explainability with accuracy, i.e., evaluating the importance of an accurate model against the value of a model that explains the motivation behind the obtained results.

The following example takes in account the MNIST (Modified National Institute of Standards and Technology) database (github.com/cvdfoundation/mnist (accessed on 28 November 2023)). It is a collection of 70,000 images consisting of handwritten digits. Each entry of the dataset is a $28 \times 28$-pixel picture depicting a digit between 0 and 9. Figure 8 reports an example of the content of the dataset. Each image has a label associated, indicating the contained digit. In particular, $X$ includes the input images, and $Y$ holds the associated labels. For example, the first image $x_0$ contains the handwritten digit 5, and therefore the value of its paired label $y_0$ is equal to 5.

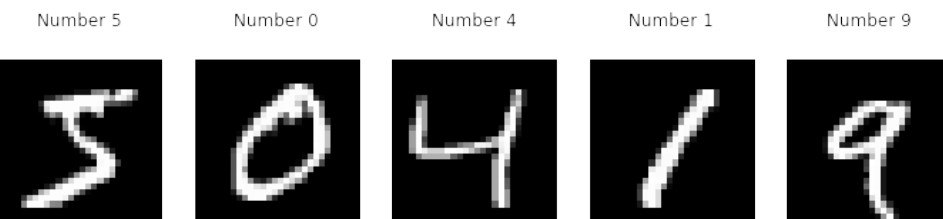

**Figure 8.** Grayscale example of the content of the MNIST dataset.

The objective is to be able to recognize the content of the image to textually reproduce it. The required classification task—leading to the creation of couples (input image, output label)—can be accomplished in multiple ways, encompassing both white-box and black-box models. For comparison, a white-box clustering algorithm (k-means) and a black-box ANN (specifically a CNN) have been selected.

Figure 9 shows the functioning of both k-means (on the left) and CNN (on the right) in the considered application.

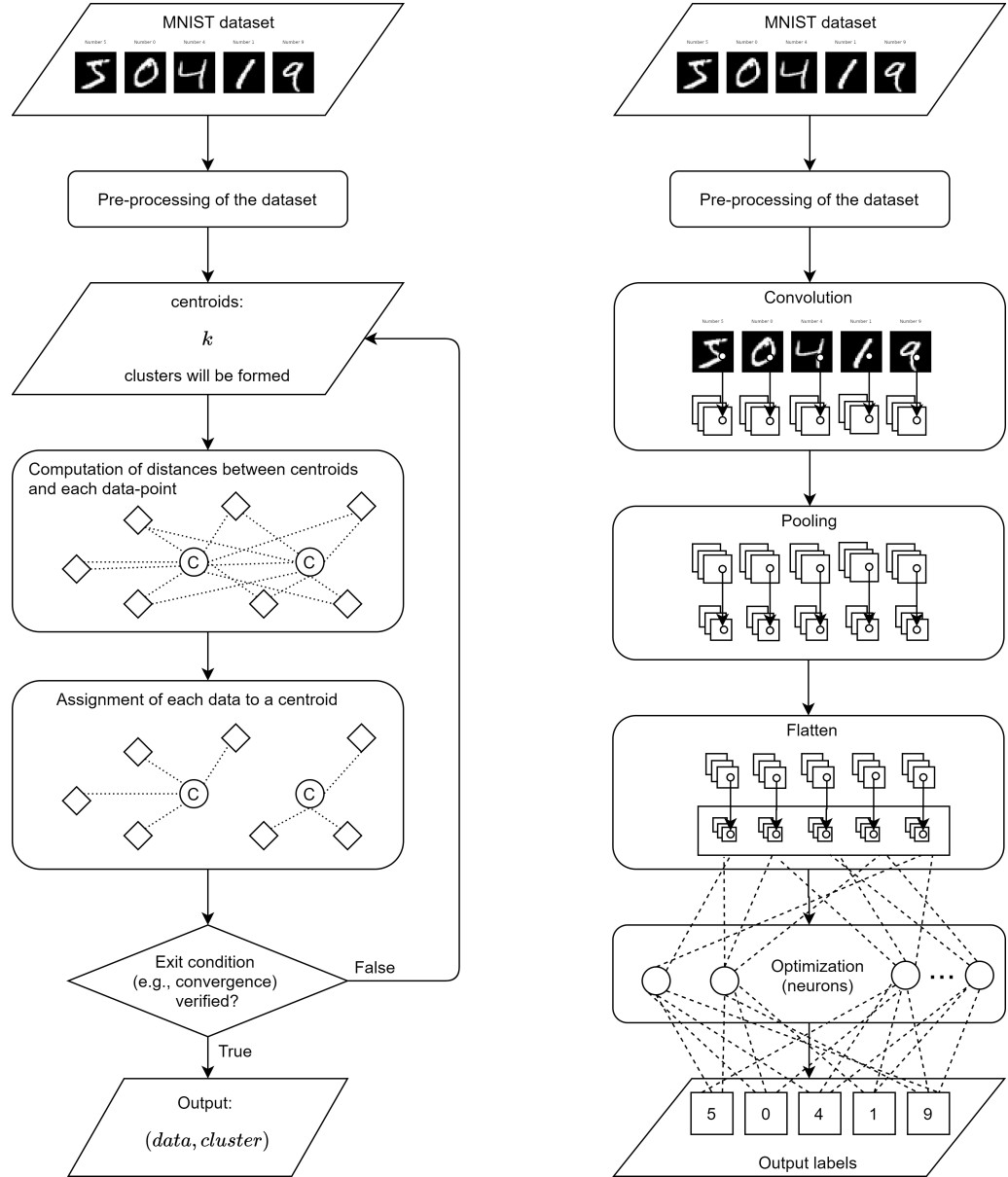

**Figure 9.** Schemas of the functioning of k-means (on the left) and CNN (on the right).

K-means algorithm is a partitioning clustering algorithm, commonly employed for grouping data points into distinct clusters based on their similarities. It exhibits clear and explicit rules to assign data points to clusters. Specifically, it provides intelligible insights into the clustering process, fostering a comprehensible and interpretable representation of its decision-making mechanisms. Moreover, the algorithm iteratively refines cluster assignments by minimizing the sum of squared distances between data points and the centroid of their respective clusters.

A CNN is a complex structure made of multiple layers and numerous parameters. Its architecture is characterized by a hierarchical arrangement of layers (including convolutional layers, pooling layers, and fully connected layers), each serving a distinct purpose (e.g., detecting patterns, pooling downsamples, dimensionality reduction, computational complexity reduction, . . . ). Fully connected layers integrate high-level features from the preceding layers, allowing the model to make predictions based on the learned representations. The relationships and patterns learned by CNNs are often not easily discernible or explainable in human terms. This lack of transparency poses challenges in under-

standing the rationale behind the network's predictions, limiting the interpretability and trustworthiness of the model in certain contexts.

After inputting the dataset for training both models, the subsequent testing phase followed. The outcome is depicted in the graph illustrated in Figure 10. Clearly, the CNN exhibited superior performance in terms of accuracy, achieving high scores in both training and testing (approaching a value of 1, the highest attainable). In contrast, K-means demonstrated only half the accuracy of CNN, indicating its comparatively lower efficacy in image classification. From an accuracy standpoint, CNN emerges as the preferable choice. However, it is worth noting that K-means provides complete transparency in its decision-making process, offering insights into how conclusions are derived, a feature not shared by CNN. As a result, the usage of a CNN enables accurate predictive power, although understanding the reason for specific predictions can be complex or even impossible.

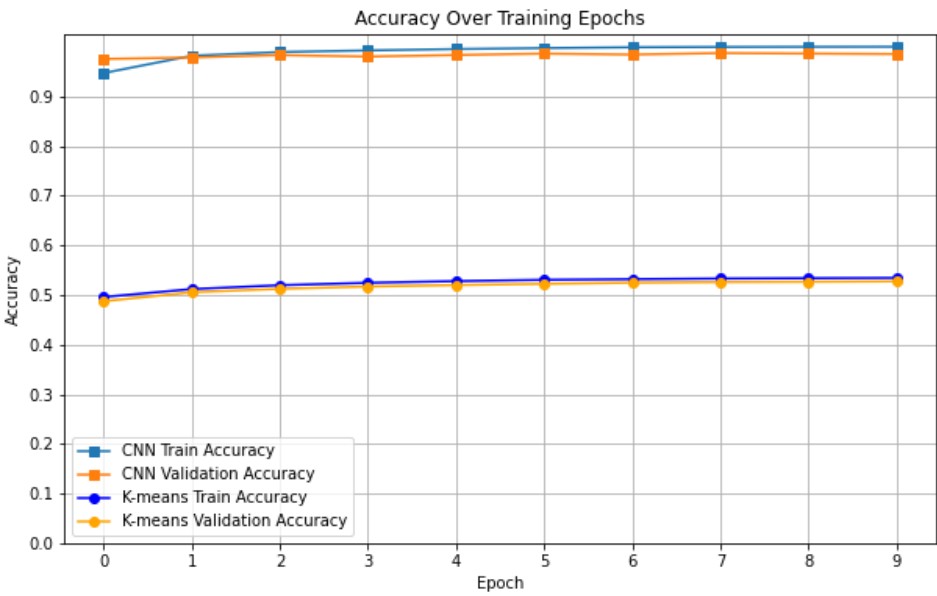

**Figure 10.** Graph showing the obtained accuracy over training epochs for both models in training and validation phases.

### 3.6. Explanation Supported by Ancillary Models

Integrating explanation frameworks with black box models in ML, such as deep neural networks, is a crucial step towards enhancing their interpretability, and one effective method to achieve this is by using decision trees. This integration primarily employs model-agnostic methods like LIME [33] or SHAP [34], which are designed to create simple, interpretable models that locally approximate the predictions of a complex model. The process involves using decision trees to mimic the decision boundary of the black box model for specific instances, thus providing a clear explanation for individual predictions. Apart from local approximations, there is also a focus on global approximations. This involves the development of simpler models like decision trees to understand the overall behavior of the complex model, a technique known as model distillation. The key challenge here lies in balancing the decision tree's simplicity to ensure interpretability while retaining enough complexity to accurately reflect the black box model's behavior. Feature importance analysis is another critical aspect, especially in ensemble methods like random forests. Decision trees can help in identifying which features the complex model deems important for making predictions. However, it is important to interpret these results cautiously, as feature importance does not necessarily imply causation. Rule extraction is another approach where decision rules are created to approximate the decision-making process of the black box model. The complexity of these rules can vary, and there is often a trade-off between the simplicity of the rules for interpretability and their complexity in capturing the nuances of the model's behavior. Ethical and practical considerations also play a significant role in

this context. Ensuring transparency and trust in AI systems, especially in sensitive areas like healthcare or finance, is paramount. Moreover, with increasing regulatory demands for explainable AI decisions, these techniques are becoming essential not just from a usability standpoint but also for compliance purposes.

In [35], the authors focus on a model-agnostic framework designed for providing explanations for vision models, evaluating various state-of-the-art methods for explaining both vision transformers (ViTs) and CNNs. Hassija et al. [36] provide a comprehensive look at different methods used in explainable AI, focusing on how to interpret black-box models across various domains, while Tan et al. [37] review global additive explanations for non-additive models, focusing on the challenges and considerations involved in explaining black-box models. In conclusion, it is possible to say that the use of decision trees to pair explanation frameworks with black box models represents an important stride towards responsible AI. This approach not only enhances the transparency of these models but also strikes a balance between their predictive accuracy and the ability to provide understandable insights into their decision-making processes. This balance is key to the responsible deployment of AI across various sectors.

### 3.7. Explainable AI in Oncology Studies

Traditionally, ML models in oncology have focused on predictive analytics, such as identifying the likelihood of a tumor being malignant based on imaging data. However, causal inference allows these models to go beyond mere correlations and delve into the underlying causes. For instance, models might discern why certain tumors respond differently to treatments, considering genetic or environmental factors. One of the most significant impacts of causal inference in ML is the advancement of personalized medicine. By understanding the causal mechanisms behind a patient's response to treatment, ML can aid in developing tailored therapies for individual patients, potentially increasing treatment effectiveness and reducing side effects. Causal models also play a crucial role in understanding disease progression. They can analyze longitudinal patient data to identify factors leading to the advancement of the disease, thus aiding in early intervention strategies. In drug development, causal inference can be a game-changer. It can identify potential new therapies or drug targets by understanding the causal pathways of cancer cells, leading to the development of more effective drugs. Moreover, causal inference can help in reducing biases in traditional ML models, which sometimes replicate biases present in the training data. By identifying and correcting these biases, the models become more accurate and fair. The transition from predictive to explanatory models in ML is pivotal for oncology. Explanatory models provide insights into why certain treatments work or why some patients have better outcomes, thus enhancing the decision-making process for clinicians. Causal inference models' ability to integrate diverse data sources, including genetic, environmental, and lifestyle factors, offers a more comprehensive understanding of cancer. This integration is crucial not only for treatment but also for healthcare policy and planning, ensuring effective resource allocation.

The recent literature witnesses significant examples in this direction, i.e., Lagemann et al. [38] explore a neural model that combines convolutional and graph neural networks. The work aims to understand causal relationships in high-dimensional biomedical problems, a critical area in oncology research. The study emphasizes the challenges and methodologies in learning causal relationships under limited data conditions, which is often the case in biological and medical research. ML can be employed [39] in classifying and predicting treatment responses in cancers of unknown primary origin. Specifically, the authors highlight the significant progress made in using genomic data and AI to improve diagnostics and treatment plans for such cancers. Featured in Synthese, [40], Buijsman's work delves into the broader aspect of causal explanations derived from ML models. While not exclusively focused on oncology, it offers valuable insights into how causal inference in ML can contribute to scientific understanding, which is essential in the context of complex diseases like cancer. Finally, Chu et al. [41] examine the progress made in the field of causal

effect estimation. It discusses the challenges faced in this domain and the potential future direction, which is highly relevant to understanding treatment effects and patient outcomes in oncology.

It has to be noticed that the integration of causal inference in ML is not just a technical improvement; it is a paradigm shift in oncology. It opens up possibilities for more effective, personalized, and informed cancer treatment and research, challenging the previous belief that ML is not suited for this field.

### 3.8. Bias and Data Fairness Issues

Data fairness in ML refers to the equitable treatment of different groups in the data used to train ML models [42]. It focuses on ensuring that the data used to train the model is representative and does not favor or disadvantage any particular group. An example of a data fairness issue can be described as a dataset used to train a hiring model containing predominantly male candidates, the model may not perform well when assessing female candidates. Achieving data fairness involves balancing and representing different groups adequately.

Given the context, the study likely explores methodologies and frameworks for identifying and mitigating biases in journalistic content. This is essential for promoting impartiality and fairness in news reporting, which is crucial for maintaining trust and credibility in media outlets. The application of data science and analytics techniques in this area represents an intersection of ML, media studies, and ethics.

Bias can affect the dataset used to train an ML model [43]. Biases in data can manifest in different forms, including (i) selection bias when certain groups may be underrepresented or excluded from the dataset, (ii) labeling bias, when errors or biases in the labeling process occur, (iii) societal bias, reflecting historical or societal prejudices against certain demographic groups; and finally, (iv) sample bias, when the dataset may not accurately represent the broader population. For example, a facial recognition system that performs poorly on certain ethnicities due to underrepresentation in the training data demonstrates bias. Another example is an employment algorithm that favors candidates from certain educational backgrounds, leading to discrimination against others. Bias in ML can be present at various stages of the development and deployment process. It often originates from biased data used to train models. If the training data contain historical biases or reflects societal inequalities, the model may learn and perpetuate these biases. Additionally, bias can be introduced through the choice of features, algorithm design, and even the labeling process of the training data. Bias can propagate through feedback loops, as biased predictions may reinforce and exacerbate existing inequalities. Bias in data is a pervasive issue, as datasets often mirror the biases and inequalities present in society. Historical and societal biases can be ingrained in various domains, including criminal justice (see, for example, the case of COMPAS [44], employment [45], healthcare [46], and finance [47]. Biased data can result from historical discrimination, cultural stereotypes, or systemic inequalities, leading to under-representation or misrepresentation of certain groups. Table 5 summarises the main types of bias.

**Table 5.** Bias classification.

| Bias | Description |
| --- | --- |
| Selection Bias | Certain groups may be underrepresented or excluded from the dataset. |
| Labeling Bias | Errors or biases in the labeling process can introduce inaccuracies. |
| Societal Bias | Reflects historical or societal prejudices against certain demographic groups. |
| Sample Bias | The dataset may not accurately represent the broader population. |

Bias in ML can significantly erode trust. When users perceive that a system produces unfair or biased outcomes, it undermines confidence in the technology. Lack of trust can lead to resistance to adopting AI solutions, particularly in critical areas such as healthcare, finance, and criminal justice. The problem is indeed serious. Unchecked bias can perpetuate

and exacerbate societal inequalities, leading to discriminatory outcomes. Moreover, biased AI systems may result in real-world harm, affecting individuals' lives and reinforcing systemic disparities. If not effectively addressed, the lack of fairness and transparency in AI systems could potentially lead to public skepticism, regulatory scrutiny, and a decline in public and institutional trust. While it might not trigger a new AI winter, it could impede the responsible and widespread adoption of AI technologies. Therefore, efforts to mitigate bias, ensure fairness, and enhance transparency in ML are critical for its sustainable and ethical development. Raza et al. [48] focuses on the critical issue of bias detection and fairness in news articles. This research is particularly relevant in the era of digital media, where news content can have a significant influence on public opinion and societal perspectives. In [49], the same authors propose, in a different work, a comprehensive AI framework that blends software engineering principles with fairness in ML, particularly for healthcare applications. The framework is designed to improve modularity, maintainability, and scalability. It begins with the identification of key actors and their roles, emphasizing the importance of understanding users to provide direction for the framework. This is followed by an analysis of requirements, where the problems in healthcare that need addressing are identified, along with specific fairness requirements, keeping in mind the ethical, legal, and social implications. Data collection is a critical step, focusing on gathering diverse and representative data samples while ensuring data privacy and security. In data pre-processing, best practices are employed to clean and normalize the data, and fairness pre-processing techniques are implemented to minimize biases. Feature selection and engineering involve identifying relevant features and using domain knowledge to create meaningful features that contribute to fairness. The model selection and training phase considers software engineering principles and employs in-processing fairness techniques during model training. The model's performance is then evaluated using both standard and fairness-specific metrics, followed by hyperparameter tuning and applying post-processing fairness techniques to refine predictions. Finally, the model is deployed and continuously monitored in a production environment, adhering to software engineering best practices. Regular evaluation of the model on new data is important to ensure its fairness and generalizability over time. User feedback is also crucial in validating the framework approach, highlighting the role of empirical science in software engineering.

### 3.9. Science and Pseudo Science

Science and pseudoscience are distinguished by their methodologies, principles, and the reliability of their claims. Precision in defining each concept is crucial for understanding their distinct characteristics. Science is a systematic and empirical endeavor aimed at acquiring knowledge about the natural world through observation, experimentation, and the formulation of testable hypotheses. It relies on a rigorous methodological approach, often referred to as the scientific method, which involves (i) Collecting data and observing natural phenomena; (ii) proposing a testable and falsifiable explanation for the observed phenomena; (iii) conducting controlled experiments to test the hypothesis; (iv) analyzing the results to draw conclusions; (v) subjecting findings to scrutiny and validation by the scientific community. The key characteristics of scientific endeavors include empirical evidence, falsifiability, repeatability, and a commitment to revising theories based on new evidence. Scientific knowledge is provisional, and subject to refinement as our understanding evolves. In contrast, pseudoscience, on the other hand, refers to activities, beliefs, or claims that are presented as scientific but lack the rigorous methodology and empirical basis of genuine scientific inquiry. Characteristics of pseudoscience include (i) lack of empirical evidence or reliance on anecdotal evidence rather than systematic observation; (ii) unfalsifiability, as pseudoscientific theories are often constructed in a way that makes them immune to falsification or disproof; (iii) pseudoscientific ideas typically lack scrutiny and validation by the broader scientific community through peer-reviewed processes; (iv) reliance on anecdotes, since pseudoscience often relies on personal testimonials or stories rather than systematic data collection; (v) persistence of belief despite contradictory

evidence and a lack of adaptability to new information. Examples of pseudoscience include astrology, homeopathy, and various paranormal claims. It is important to note that the term "pseudoscience" does not necessarily imply intentional deception; individuals may sincerely believe in pseudoscientific ideas due to cognitive biases or a lack of understanding of scientific principles.

While a biased ML model may produce outputs that reflect or even amplify existing biases (such as racism and sexism), it would be an oversimplification to categorize such a model as pseudoscience. Biases in ML models can be identified and mitigated through careful analysis, reevaluation of data, and improvements to algorithms. They are contextual and can be traced back to the data and processes involved. Although ML is not a traditional natural science in the sense of physics or chemistry, it shares commonalities with scientific inquiry. It involves hypothesis testing, experimentation, and validation processes, although the emphasis is often on predictive accuracy rather than causal explanations.

### 3.10. RL, IL, and Human Learning

RL and human learning based on punishment and compensation share similarities in their fundamental principles of learning from feedback through a system of rewards and penalties.

RL algorithms operate on a feedback loop where an agent receives feedback in the form of rewards or penalties based on its actions. The algorithm learns to optimize its behavior to maximize cumulative rewards over time. Humans often learn through a similar feedback mechanism. Positive experiences or outcomes are akin to rewards, encouraging the repetition of certain behaviors, while negative experiences or consequences act as punishments, discouraging undesirable actions. On the other hand, RL involves the challenge of assigning credit to past actions that contribute to current outcomes. The algorithm must understand which actions led to rewards or penalties, even if they occurred temporally distant from the outcome. Similarly, humans often need to attribute consequences, whether positive or negative, to specific actions taken in the past. This understanding informs future decision making to repeat successful actions or avoid those associated with negative outcomes. Rather interestingly, RL models often need to generalize knowledge gained in one context to perform well in new, unseen situations, opposite to humans exhibiting a capacity for generalization, applying lessons learned from one domain to make informed decisions in novel or related situations.

A similar parallelism could be established between IL and human learning. Mirror neurons represent specialized neuronal entities manifesting a distinctive property. Their activation occurs not only during the execution of a particular action by an individual but also upon witnessing a comparable action being performed by another individual. This distinctive dual-response characteristic implies a pivotal role of mirror neurons in the comprehension and interpretation of the actions of others, thereby facilitating empathic resonance with their emotional states and intentions. The identification of mirror neurons within the macaque premotor cortex (PMC) by Rizzolatti et al. [50,51] in the early 1990s revolutionized the landscape of our comprehension regarding the neural substrates of social cognition. Subsequent investigations have substantiated the presence of mirror neurons across diverse cerebral regions, including the inferior frontal gyrus (IFG), supplementary motor area (SMA), and insula. The activation of mirror neurons during the observation of actions is posited to emanate from a phenomenon termed "motor resonance". This mechanism entails the simulation of observed actions within the human motor system, mirroring the neural activity analogous to that which would be engendered if we were executing the observed action. This simulation process engenders an understanding of the intention and objective underlying the observed action, thereby facilitating empathetic responses and social interactions. Mirror neurons have been implicated in an expansive array of social cognitive processes, encompassing different research fields. The role of mirror neurons supports different interpretations [52]. For example, they empower the comprehension of the significance and purpose of observed actions, affording the ability to

anticipate the likely outcomes of such actions. Likewise, they facilitate the acquisition of novel skills and behaviors through observational learning, enabling individuals to emulate and adopt actions demonstrated by others. Mirror neurons substantially contribute to the empathetic experience by facilitating the shared emotional resonance with others, fostering an enhanced understanding of their emotions and perspectives. Furthermore, they could play a role in the theory of mind, the capacity to attribute mental states to oneself and others, thereby enabling an understanding of the beliefs, intentions, and desires of fellow individuals.

Indeed, there are notable parallels between IL in the ML field and imitation learning in cognitive science, despite the different domains and mechanisms involved (see [53]). Both fields leverage the concept of learning from observed behavior, albeit with distinct methodologies and objectives. Firstly, IL involves training a model by observing demonstrations of a task performed by an expert. The model learns to imitate the expert's actions and behavior. Similarly, individuals observe and mimic the actions of others as part of the social learning process. Secondly, the transfer of knowledge applies to both realms. IL allows knowledge transfer from a knowledgeable agent (demonstrator) to a learning agent, enabling the latter to acquire skills without explicitly programmed rules. On the other hand, imitation in cognitive development allows individuals to transfer knowledge and acquire new skills by observing and replicating the behaviors of others in their social environment. Thirdly, IL often involves implicit learning, where the learning algorithm extracts patterns and strategies from observed demonstrations without explicit instruction. At the same time, children—for example—engage in implicit imitation learning as they observe and replicate actions in their environment without explicit teaching. However, there are other similarities. For example, IL models can adapt to variations in the environment and generalize their learned behaviors to new, unseen situations, as opposed to individuals, who can adapt learned behaviors to different contexts and generalize skills acquired through imitation to various situations. IL is also crucial in the development of socially intelligent agents, allowing them to interact effectively with humans and other agents and respond well against challenges in ambiguous or complex environments where the mapping between actions and outcomes is not straightforward.

## 4. Related Work

This section embraces different perspectives on the methodological aspects of ML. It is articulated in three subsections, reviewing (i) the perspective of ML as agnostic science, (ii) the urge for the validation of the scientific method employed by ML, and finally (iii) the successful contributions to ML as a science within other disciplines. A dedicated subsection reviews the originality of the present contribution concerning the scientific literature.

### 4.1. ML as Agnostic Science

The problem of interpretability has become more and more urgent in the realm of ML, although a clear definition of the term is far from clear. For example, Krishnan [54] claims that interpretation within the context of ML systems refers to the elucidation or presentation of information in comprehensible terms. In the field, interpretability is formally defined as the capacity to explicate or present information in a manner understandable to human comprehension. Within a rigorous framework, model transparency is characterized by the feasibility for an individual to contemplate the entirety of the model simultaneously, as proposed by Lipton et al. [55], with a specific emphasis on the all-encompassing nature of this transparency. A secondary aspect of transparency involves the accessibility of intuitive explanations for each component of the model, including inputs, parameters, and calculations. In the representation referred to as "transparent", the states of a system are portrayed in a manner amenable to explicit scrutiny, analysis, interpretation, and understanding by humans. Moreover, transitions between these states are delineated by rules possessing analogous interpretative properties.

Napoletani et al. [56] reconsider the role of computational mathematics has changed in light of new disciplines, such as genomics, big data techniques, and data science. The authors introduce four methodologies to clarify the approach to ML, i.e., (i) the microarray paradigm, (ii) the pre-eminence of historical phenomena, (iii) the conceptualization of developmental processes, and finally, (iv) the principle of forcing. Concerning the microarray paradigm, researchers believe that a high volume of data about a phenomenon, opportunely queried, can provide meaningful insight into the phenomenon itself. The authors remark that in the microarray paradigm, solutions are derived through an automated process of fitting data to models devoid of any inherent structural comprehension beyond the immediate resolution of the problem. In this sense, they formulate the theory of "agnostic science". The pre-eminence of historical phenomena is supported by the concept of fitness landscape (a metaphorical representation of the relationship between the genetic makeup of organisms and their ability to survive and reproduce in a specific environment), allowing the authors to infer that the timeframe for transitioning from one local optimization process to another in the development of such phenomena is considerably longer than the time required for the optimization processes themselves. According to the authors, the conceptualization of developmental processes can be understood by considering the theory of evolution, epitomizing how a qualitative framework can engender quantitative understanding. Despite its non-mathematical nature and lack of quantitative predictions, the theory's inherent logical structure and conceptual framework have served as a cornerstone of biological inquiry, fostering the development of specific mathematical models since its inception. Finally, the principle of forcing consists of the notion that various distinct techniques, which have been developed to apply advanced mathematical concepts to empirical problems, can be unified under a shared methodological perspective. This perspective revolves around the idea of systematically applying mathematical ideas and methods to the data. As a result, according to Napolitani et al., the concept of historical phenomena and the principle of forcing appear relevant primarily within the context of extensive data sets, thus exhibiting a profound reliance on the microarray paradigm. The articulation of historical phenomena inherently redirects focus from the states of phenomena towards the developmental processes that give rise to them. The motif inspired by the principle of forcing is delved into by Napolitani et al. [57]. The authors contend that numerous optimization methods can be construed as exemplifying the concept of forcing. This occurs even in instances where detailed and credible models of a phenomenon are absent or fail to substantiate the application of said technique. Specifically, the implication of forcing is demonstrated in particle swarm optimization methods and the optimization-based modeling of image processing problems. From these observations, a principle applicable to general data analysis methods is extrapolated, termed 'Brandt's principle.' This principle posits that an algorithm achieving a steady state in its output has effectively resolved a problem or necessitates replacement. Finally, it is posited that biological systems, and phenomena adhering to general morphogenetic principles, represent a natural context for the application of this principle.

The theory of data analysis as "agnostic science" is further explored by Napoletani et al. [58,59] posits that data science constitutes a cohesive and innovative approach to empirical challenges that, in its broadest sense, does not contribute to the comprehension of phenomena. Within the novel mathematization inherent in data science, mathematical techniques are not chosen based on their relevance to a particular problem; rather, they are employed through a process of 'forcing.' This involves the application of mathematical methods to a specific problem solely based on their capacity to reorganize data for subsequent analysis and the inherent richness of their mathematical structure. Specifically, we contend that a comprehensive understanding of DL neural networks emerges within the framework of forcing optimization methods. Lastly, we delve into the broader inquiry concerning the suitability of data science methodologies in problem solving. Our argument emphasizes that this inquiry should not be construed as a quest for a correlation between phenomena and specific solutions generated by data science methods. Instead, the focus

is on elucidating the internal structure of data science methods through precise forms of understanding.

### 4.2. Discussions around the Validity of a Scientific Method in ML

The methodological approaches in ML research can be reviewed more precisely. For example, ref. [60] highlights a gap between current ML practices and the traditional scientific method, particularly in the areas of hypothesis formulation and statistical testing. The authors suggest that ML research often lacks the systematic rigor found in other scientific disciplines. They advocate for incorporating empirical science methodologies into ML, including controlled, reproducible, and verifiable experimental designs. This, they argue, would not only improve the foundational science of DL but also enhance the quality and reliability of applied ML research. From a different angle, Krenn et al. [61] explore how artificial intelligence (AI) can contribute to scientific understanding. It presents a philosophy of science framework to evaluate AI's role in scientific understanding and identifies three dimensions where AI can contribute: as a computational microscope, as a resource of inspiration, and as an agent of understanding. Each dimension is examined through various examples, highlighting AI's potential to reveal new scientific insights, inspire novel ideas, and potentially develop new scientific understanding autonomously. The article emphasizes the distinction between scientific discovery and understanding, underscoring AI's role in advancing scientific knowledge.

Van Calster et al. [62] discuss how the calibration of predictive algorithms in clinical decision making is critical. Inaccurate calibrations can lead to false expectations and misguided decisions, emphasizing the need for predictive models that are well-calibrated for the specific population and setting. The heterogeneity of patient populations and changing healthcare dynamics necessitate continuous monitoring and updating of models. The overarching goal is to enhance the efficacy of predictive analytics in shared decision making and patient counseling.

Varoquax et al. [63] provide a comprehensive set of guidelines for effectively implementing and evaluating ML in the area of medical imaging. This work stresses the critical need for data integrity, emphasizing the separation of test data right from the outset to avoid data leakage. The document also underscores the necessity of having a clearly defined methodology for choosing model hyperparameters, while avoiding the use of test data in this process. The authors further advise on the importance of having a sufficiently large test dataset to ensure statistical significance, recommending hundreds or ideally thousands of samples, with performance metrics supported by confidence intervals. This paper advocates for the use of diverse data sets that accurately reflect patient and disease heterogeneity across multiple institutions, incorporating a wide range of demographics and disease states. The authors call for strong baseline comparisons that include not only state-of-the-art ML techniques but also traditional clinical methods. They also highlight the need for a critical discussion on the variability of results, considering random elements and data sources. Moreover, it is suggested to employ a variety of quantitative metrics to capture different aspects of the clinical problem, linking them to pertinent clinical performance indicators, and making informed decisions about trade-offs between false detections and misses. Additionally, it is recommended to include qualitative insights and involve groups most impacted by the application in the design of evaluation metrics.

Bouthillier et al. [64] review the balance between empirical and exploratory research in the field of ML, particularly in DL. The authors highlight the importance of empirical research in building a robust knowledge base and ensuring steady scientific progress. Conversely, exploratory research is valued for its role in expanding the research horizon with new findings. However, it is important to pay attention to overly focusing on either approach, as it can lead to non-robust foundations or hinder progress by limiting exploration. Recent critiques in DL methodology are mentioned, emphasizing the need to understand the balance between these two research methods. The authors use the example of batch-norm in DL to illustrate the importance of both exploratory and empirical research,

suggesting that a better synergy between the two can lead to more efficient and rational development in the field.

### 4.3. ML as a Science

Thiyagalingam et al. [65] support the view that ML can be regarded as a science by emphasizing the importance of benchmarking in the field. It discusses the challenges and approaches in developing benchmarks for scientific ML, highlighting how these benchmarks are essential for evaluating and improving ML algorithms in scientific applications. The paper underscores the need for systematic, reproducible, and verifiable methods to assess ML techniques, which are key principles in scientific research. This approach to benchmarking in ML aligns with the scientific method, suggesting rigorous testing, evaluation, and improvement of algorithms based on empirical data and controlled experiments. In [66], Researchers are encouraged to explore the use of AI applications in mathematics education, focusing on providing personalized guidance to students and examining the effects of AI-based learning methods. An innovative approach involves using educational data mining (EDM) to explore factors that influence student outcomes and the relationship between student behaviors and performance. Implementing AI in advanced mathematics courses, such as geometry, topology, and applied mathematics, as well as in interdisciplinary programs like STEM, could be highly beneficial. There is also a need to understand how AI can assist underrepresented groups in math education, including teachers and high school students. Beyond quantitative analysis, qualitative research methods are important for gathering student feedback on AI-assisted learning and delving into their perceptions. Developing adaptive learning environments in mathematics through collaboration between education, educational technology, and computer science experts is another key area. Incorporating modern AI technologies like DL, which offer tools like image and voice recognition, can provide unique benefits, such as aiding visually impaired students. The incorporation of AI in mathematics education underscores its scientific nature. This approach involves systematic research and experimentation, fundamental to the scientific method. Educational data mining (EDM) emphasizes AI's role in data analysis, a key scientific process. AI's interdisciplinary application in advanced mathematics and STEM courses highlights its integration with other scientific fields, illustrating its scientific versatility. The development of adaptive learning environments through collaboration among experts across multiple disciplines showcases AI as both a beneficiary and contributor to scientific knowledge. The use of advanced technologies like DL and image recognition in education reflects ongoing scientific innovation within AI. Furthermore, employing both qualitative and quantitative research methods in studying AI's impact demonstrates a commitment to scientific rigor. Finally, AI's capacity to address complex educational challenges, like cognitive load and learning anxiety, exemplifies its role in scientific problem solving. Collectively, these aspects demonstrate that AI is deeply rooted in scientific principles and methodologies, confirming its status as a robust scientific discipline.

Douglas et al. [67] discusses the application of ML in theoretical science, illustrating how ML methods, including those relying on synthetic data, are being used in fields like mathematics and theoretical physics. This integration of ML into traditional scientific domains demonstrates its scientific credibility and its potential to advance various fields of study. Ourmazd et al. [68] explores the impact of ML in the scientific domain. It presents various examples where ML has been applied to understand complex phenomena, such as analyzing noisy data and studying the conformational landscape of viruses. This illustrates how ML is not just a tool but a scientific approach that contributes to deeper understanding and discovery in science.

### 4.4. Contributions of This Work

This work presents, concerning the existing literature, a critical review of the foundations of ML, focusing on its reliance on inductive reasoning and questioning the field's capacity for genuine learning. The article uniquely critiques the traditional acceptance of

inductive reasoning in ML, highlighting philosophical concerns about the reliability and validity of conclusions drawn from specific instances to general rules. This stance aligns with philosophical discussions about the 'problem of induction', challenging the fundamental premise that high accuracy in pattern recognition equates to true learning. Additionally, this article delves into the nuanced differentiation between the old and new aspects of ML. It provides a critical historical analysis, distinguishing the statistical underpinnings of methods like supervised learning from more recent developments like convolutional neural networks (CNNs), long short-term memory networks (LSTMs), and large language models (LLMs). This approach contributes to a deeper understanding of ML's evolution, shedding light on how contemporary advancements either build upon or diverge from traditional methods. Furthermore, this work's exploration of what constitutes novelty in ML techniques contributes to the ongoing debate about the nature of innovation in the field. It offers a grounded perspective that scrutinizes whether recent developments in ML represent genuine breakthroughs or are merely incremental advancements or adaptations of existing techniques from other disciplines. Overall, this article adopts a critical and philosophical lens, less commonly seen in mainstream ML literature, which often emphasizes technological and algorithmic progress. This perspective is valuable as it encourages researchers and practitioners to critically assess ML's capabilities and development trajectory.

## 5. Conclusions

Diversely from AI, which suffered repeated winters (i.e., periods where the credibility of the discipline dramatically decreased to the point of suspending research funds), ML is here to stay. However, different challenges make its path not always linear. Firstly, its foundational epistemological approach—induction—is not entirely correct in producing valid inferences. Secondly, some ML models suffer from the lack of causal explanation mechanisms, making it difficult to assess their scientific status and affecting the trust of specialized users (such as oncologists or market traders). Thirdly, issues such as bias and data fairness greatly affect ML models, undermining the cultural and social perception of ML and, at large, of AI. However, these faults can be mitigated by methods and techniques proposed by the scientific literature. For example, explanation frameworks can be paired with "black models", using decision trees or polynomial regressors. Similarly, the presence of bias can be detected and mitigated by pre-processing, in-processing and post-processing algorithms. More interestingly, techniques such as RL and IL seem to suggest a stronger similarity in the way ML models and humans learn, possibly implying a more robust theoretical foundation of AI in general.

**Author Contributions:** Investigation, A.G.; Writing—original draft, E.B. and A.G.; Writing—review & editing, E.B. All authors have read and agreed to the published version of the manuscript.

**Funding:** This research received no external funding.

**Data Availability Statement:** Not applicable.

**Conflicts of Interest:** The authors declare no conflicts of interest.

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
