# Peer review of "The Challenges of Machine Learning: A Critical Review"

_electronics, doi:10.3390/electronics13020416_

Round 1

Reviewer 1 Report

Comments and Suggestions for Authors

This paper argues against the widespread belief that Machine Learning (ML) is a science. The authors contend that while ML has a solid mathematical and statistical foundation, it cannot be considered a strict science. Moreover, the paper argues against the commonly accepted notion that a machine can learn from data by merely applying statistical methods in a supervised or unsupervised mode. 

The reviewer has also the following comments:

1- The difference between Figures 5 and 6 is not clear.

2- In Table 2, how do "Intrinsic Structures" represent feedback in UL? 

3- In Table 2, policy learning is also a key mechanism for RL.

4- In Line 289,  what is the reference for "Lipton in 2018"?

5- In the Related Works section, there are no clear comparisons with other works that either favor ML's idea of learning/science or against it. The author should include other papers that present these points of view in this section. 

6- Although in Line 264, the authors claim that "The argument concerning ML as science does not hold entirely," they say in Line 371 that "Machine Learning can be characterized as a cutting-edge science," which the reviewer finds a contradiction. The authors should clarify their stand and claims.  

7- The paragraph starting from Line 422 needs clarification.

8- In Line 542, XAI is presented without mentioning what it stands for.

9- The Discussion section should be enriched by a realistic example that contains a real dataset or datasets and apply one or more ML techniques (like ANN) to them. This will better highlight the ideas of the authors over the different subsections of this section.

Comments on the Quality of English Language

The paper needs proofreading as there are some English mistakes and typos like "hibrid".

In Line 256,  there should be a space after the period.

In Line 275, "Ln" should read "In".

Author Response

The authors would like to extend their sincere gratitude for the time and effort invested by the reviewer in commenting on this article. 
The insightful comments and constructive feedback have been invaluable, significantly enriching the content and depth of this work. 
The reviewer's perspectives have not only helped the authors to clarify and strengthen the arguments presented but have also added notable value to the overall discourse of the article. Please find in the following the answers to the reviewer's questions.

1- The difference between Figures 5 and 6 is not clear.

[Answer: All the figures have been fully edited, and the difference between figure figures 5 and 6 has been enhanced to be  clearer.]

2- In Table 2, how do "Intrinsic Structures" represent feedback in UL? 

[Answer: "Intrinsic Structures" has been replaced by "Evaluation Metrics or Criteria" as the algorithm tries to identify patterns, relationships, or structures in the data without specific guidance on what the output should be. The learning process is driven by the inherent properties of the data rather than by a predefined target. In unsupervised learning, evaluation metrics or criteria are often used to assess the quality of the model's output, but these are not exactly "feedback signals" in the same sense as supervised learning.]

3- In Table 2, policy learning is also a key mechanism for RL.

[Answer: Policy learning topic has been added.]

4- In Line 289,  what is the reference for "Lipton in 2018"?

[Answer: The reference for "Lipton 2018" has been added.]

5- In the Related Works section, there are no clear comparisons with other works that either favor ML's idea of learning/science or against it. The author should include other papers that present these points of view in this section. 

[Answer: The Related Work section - which has been shifted after the discussion part, to be more coherent - has been greatly enhanced by introducing different subsections: i) ML as agnostic science, ii) Discussions around the validity of a scientific method in ML, which urges caution in claiming that ML embeds a strong scientific method , and iii) ML as a science, supporting instead opposite view

6- Although in Line 264, the authors claim that "The argument concerning ML as science does not hold entirely," they say in Line 371 that "Machine Learning can be characterized as a cutting-edge science," which the reviewer finds a contradiction. The authors should clarify their stand and claims.  

[Answer: Line 371 that "Machine Learning can be characterized as a cutting-edge science," which the reviewer finds a contradiction has been removed being confusing.]

7- The paragraph starting from Line 422 needs clarification.

[Answer: The paragraph has been rewritten to add clarity.]

8- In Line 542, XAI is presented without mentioning what it stands for.

[Answer: The XAI term has now been written in full to avoid any misunderstanding.]

9- The Discussion section should be enriched by a realistic example that contains a real dataset or datasets and apply one or more ML techniques (like ANN) to them. This will better highlight the ideas of the authors over the different subsections of this section.]

[Answer: A simple example of k-means and CNN algorithms on a real dataset (MNIST) has been added in order to discuss the interesting trade-off between accuracy and explanatory power of ML models.]

Comments on the Quality of English Language

The paper needs proofreading as there are some English mistakes and typos like "hibrid".

In Line 256,  there should be a space after the period.

In Line 275, "Ln" should read "In".

[Answer: All the typos have been checked and fixed. The language has been verified and improved.]

Reviewer 2 Report

Comments and Suggestions for Authors

This paper explores challenges associated with different forms of ML. There is in-depth explanation of ML types. There is discussion on interpretability of ML methods.  Some history on evolution of ML algorithms have also been discussed. The following major concerns should be addressed prior to publication. 

1. Causal inference of ML based predictions is now being studied extensively. As such the claim that ML cannot be used for oncology can be reconsidered with examples. 

2. Building on point 1, this research does not explore a lot of recent developments in AI, ML, and ANN. There should be more technical discussion on these methods and how they can be used for prediction with high confidence levels. 

3. Minor grammatical errors: Line 221 should be 'hybrid'. 

4. Related work section lacks depth as most of the discussion circulates around a couple of papers. This needs heavy rework. 

5. Lines 406-413 needs references. 

6. Citation issue in Line 587. 

7. Most of the topics in the discussion section is comprised of factors which the community is already aware of like bias, fairness, explainable AI, etc. As such, what is the significance of this section besides reiterating these known factors? 

8. A discussion on different types of AI focused research done in these domains along with their advantages/limitations would significantly increase the impact of the discussion section. 

9. Provide discussion on how explanation frameworks can be paired with black models, using decision trees. 

10. Some discussions can be condensed. For example, page 14 on induction. Most of the explanation does not relate to AI and ML. 

11. To summarize, as its a review paper, please explore more research to explain the challenges in ML. 

Author Response

The authors would like to extend their sincere gratitude for the time and effort invested by the reviewer in commenting on this article.
The insightful comments and constructive feedback have been invaluable, significantly enriching the content and depth of this work.
The reviewer's perspectives have not only helped the authors to clarify and strengthen the arguments presented but have also added notable value to the overall discourse of the article. 
Please find in the following the answers to the reviewer's questions.

1. Causal inference of ML based predictions is now being studied extensively. As such the claim that ML cannot be used for oncology can be reconsidered with examples. 

[Answer: The Discussion section has been enriched with a subsection concerning Explainable AI in oncology studies.]

2. Building on point 1, this research does not explore a lot of recent developments in AI, ML, and ANN. There should be more technical discussion on these methods and how they can be used for prediction with high confidence levels. 

[Answer: a discussion related to the accuracy of prediction on ML has been added, including methods such as ensemble methods, hyperparameter optimization, grid search, random search, bayesian optimization, transfer learning, and regularization.]

3. Minor grammatical errors: Line 221 should be 'hybrid'. 

[Answer: The errors in the text have been corrected.]

4. Related work section lacks depth as most of the discussion circulates around a couple of papers. This needs heavy rework. 

[Answer: The Related Work has been fully re-edited, by subdividing the work in 3 areas: ML as agnostic science, articles supporting the urge of a solid scientific method within ML and finally, a more positive view of ML as a science, contributing in enhancing other scientific displines.]

5. Lines 406-413 needs references. 

[Answer: The text has been enriched with proper scientific references.]

6. Citation issue in Line 587. 

[Answer: The citation has been corrected.]

7. Most of the topics in the discussion section is comprised of factors which the community is already aware of like bias, fairness, explainable AI, etc. As such, what is the significance of this section besides reiterating these known factors? 

[Answer: Although the issues of bias, fairness, and explainable AI are well-known in the ML community,  their implications and challenges evolve with advancements in technology and applications. This section revisits these issues in light of recent developments, new research findings, and emerging technologies, showing how these perennial concerns are manifesting.
Furthermore, it is necessary to stress the urgency of continuous discussion about these issues in the ML community. The fact that these issues are well-known does not diminish their importance; ongoing dialogue is crucial for developing better practices, updating policies, and educating newcomers to the field. 
In this sense, this section makes these issues accessible or relevant to a broader, perhaps non-specialist audience. The implications of bias, fairness, and explainability in ML extend beyond the technical community to sectors like policy-making, legal affairs, and public perception. Furthermore, it is important to clarify the epistemological position of ML, what is novel, and what is derived from other disciplines.]

8. A discussion on different types of AI focused research done in these domains along with their advantages/limitations would significantly increase the impact of the discussion section. 

[Answer: In the Discussion section, the "Bias and data fairness issue" subsection has been enriched with a discussion about ML techniques to mitigate these issues.]

9. Provide discussion on how explanation frameworks can be paired with black models, using decision trees. 

[Answer: a discussion about explanation frameworks can be paired with black models using decision trees has been added in the new subsection "Explanation supported by ancillary models".]

10. Some discussions can be condensed. For example, page 14 on induction. Most of the explanation does not relate to AI and ML. 

[Answer: Although this point is essentially correct for ML researchers and Computer Scientists in general, this article aims to reach a broader audience, including for example philosophers of science and epistemologists. The latter initiated around the 80s a debate about induction as a foundation of ML, identifying in this point a general methodological weakness.]

11. To summarize, as its a review paper, please explore more research to explain the challenges in ML.

[Answer: the paper has been heavily edited, adding new sections in the Related Work and Discussion sections. A new section about the different types of ML has been added as well.]

Reviewer 3 Report

Comments and Suggestions for Authors

Reviewer comments on the paper titled "The Challenges of Machine Learning. A critical review”

Comments:
The authors present a hybrid algorithm that the Challenges of Machine Learning. A critical review. My comments are as follows:

There are many issues in the assertion and argument section. A few things are listed below. Before accepting, please fix all the issues in the assertion and argument section.

1.     Overall, the abstract has the necessary details, but some abbreviations used are not first defined (eg. CT-scan image,), which makes it hard to understand the proposed algorithm.

2.     Table 1, on page 2 shows some ML model learning benchmarks, the authors should cover more machine learning model benchmarks.

3.     Table 3. shows the ML model in chronological order but the Table contains some classification/regression algorithms, the authors should provide the chronological order for both classification/regression and clustering algorithms such as

·      k-Means

  • k-Medians
  • Expectation Maximization (EM)
  • Hierarchical Clustering

4.     Regardless of the scientific content of this manuscript, it is full of typographical, grammatical, and organizational errors.

5.      Grammar and time tenses of verbs should be revised.

6.     Each equation in the manuscript should be given an equation number such as the equation in lines 204 and 207.

7.     To present the author's work, it is better to support their argument experimentally with benchmark datasets.

8.     For each figure in the manuscript the authors should cite the original source.

Comments on the Quality of English Language

   1 - Regardless of the scientific content of this manuscript, it is full of typographical, grammatical, and organizational errors.   

     2 - Grammar and time tenses of verbs should be revised.

Author Response

The authors would like to extend their sincere gratitude for the time and effort invested by the reviewer in commenting on this article.
The insightful comments and constructive feedback have been invaluable, significantly enriching the content and depth of this work.
The reviewer's perspectives have not only helped the authors to clarify and strengthen the arguments presented but have also added notable value to the overall discourse of the article. 
Please find in the following the answers to the reviewer's questions.

1. Overall, the abstract has the necessary details, but some abbreviations used are not first defined (eg. CT-scan image,), which makes it hard to understand the proposed algorithm.

[Answer: The abstract has been edited removing any acronym.]

2. Table 1, on page 2 shows some ML model learning benchmarks, the authors should cover more machine learning model benchmarks.

[Answer: The table has been enhanced with more benchmarks.]

Table 3. shows the ML model in chronological order but the Table contains some classification/regression algorithms, the authors should provide the chronological order for both classification/regression and clustering algorithms such as

 k-Means

k-Medians

Expectation Maximization (EM)

Hierarchical Clustering

[Answer: The table has been enriched with more models covering areas different from the  classification/regression topics.]

3. Regardless of the scientific content of this manuscript, it is full of typographical, grammatical, and organizational errors.

[Answer: The text has been edited to remove grammar errors and typos.]

4. Grammar and time tenses of verbs should be revised.

[Answer: Grammar error and time tenses have now been revised.]

5. Each equation in the manuscript should be given an equation number such as the equation in lines 204 and 207.

[Answer: All the equations have now been numbered.]

To present the author's work, it is better to support their argument experimentally with benchmark datasets.

[Answer: The authors feel that providing an extensive study including benchmark details would deviate from the original scope of the article. At the same time, the authors would like to take into due consideration the reviewer's advice and plan a separate work dedicated to this task in the future.[

6. For each figure in the manuscript the authors should cite the original source.

[Answer: all the images have been re-edited by the authors.]

Comments on the Quality of English Language

   1 - Regardless of the scientific content of this manuscript, it is full of typographical, grammatical, and organizational errors.   

[Answer: The text has been edited to improve the quality of English grammar.]

     2 - Grammar and time tenses of verbs should be revised.

[Answer: The use of the verbs have been verified and corrected.]

Reviewer 4 Report

Comments and Suggestions for Authors

The paper presents a compelling and thought-provoking study. However, I would like to raise certain concerns about its novelty. Despite these concerns, I appreciate the value of review articles in providing comprehensive insights. Below are some specific suggestions for improving the paper:

The introductory section of the paper is quite lengthy and could benefit from being divided into smaller, more focused sections. This approach would help in better organizing the content and make it easier for readers to grasp the main ideas.

To streamline the introduction, consider moving some of the introductory content into a "Related Work" section or creating a dedicated section titled "Overview of Traditional Approaches" (feel free to put bette header). This would help provide context and set the stage for the paper's main contributions more effectively.

Incorporating a mindmap figure into the introduction is an excellent idea. Such a visual representation can offer readers a clear and concise overview of the paper's structure, helping them understand the logical flow of the content right from the beginning. It's imp for the authors to explicitly articulate the unique contribution of their work. This can be achieved through a dedicated section where they summarize their key findings and emphasize what sets their research apart from existing literature. How the authors work differnt from previous works.

Acknowledging the relevance of Large Language Models (LLMs) in advanced ML methods is crucial, even if it's not the primary focus of the paper. Consider including a brief section that introduces and discusses the role of LLMs in the context of the study's subject matter.

 To strengthen the discussion on bias, it is indeed beneficial to cite relevant works. Suggesting specific citations, https://link.springer.com/article/10.1007/s41060-022-00359-4 and https://ojs.aaai.org/index.php/AAAI-SS/article/view/27493, can help readers explore the topic of bias in ML more deeply.

Authors need to highlight their unique contributions in the very beginning.

Comments on the Quality of English Language

can be improved, break longer paras into smallers 

Author Response

The authors would like to extend their sincere gratitude for the time and effort invested by the reviewer in commenting on this article.
The insightful comments and constructive feedback have been invaluable, significantly enriching the content and depth of this work.
The reviewer's perspectives have not only helped the authors to clarify and strengthen the arguments presented but have also added notable value to the overall discourse of the article. 
Please find in the following the answers to the reviewer's questions.

1. The introductory section of the paper is quite lengthy and could benefit from being divided into smaller, more focused sections. This approach would help in better organizing the content and make it easier for readers to grasp the main ideas.

[Answer: The Introduction part is indeed lengthy and it has been decomposed into two sections.]

2. To streamline the introduction, consider moving some of the introductory content into a "Related Work" section or creating a dedicated section titled "Overview of Traditional Approaches" (feel free to put bette header). This would help provide context and set the stage for the paper's main contributions more effectively.

[Answer: The "Overview" section has been added with the title "Machine Learning Paradigms".]

3. Incorporating a mindmap figure into the introduction is an excellent idea. Such a visual representation can offer readers a clear and concise overview of the paper's structure, helping them understand the logical flow of the content right from the beginning.

[Answer: A mindmap in the shape of a block diagram has been added in the introduction section.]

4. It's imp for the authors to explicitly articulate the unique contribution of their work. This can be achieved through a dedicated section where they summarize their key findings and emphasize what sets their research apart from existing literature. How the authors work differnt from previous works.

[Answer: a subsection of the Related Work addresses this concern.]

5. Acknowledging the relevance of Large Language Models (LLMs) in advanced ML methods is crucial, even if it's not the primary focus of the paper. Consider including a brief section that introduces and discusses the role of LLMs in the context of the study's subject matter.

[Answer: The LLMs are now discussed in a dedicated subsection. This subsection is part of a new section reviewing the principal ML paradigms.]

6. To strengthen the discussion on bias, it is indeed beneficial to cite relevant works. Suggesting specific citations, https://link.springer.com/article/10.1007/s41060-022-00359-4 and https://ojs.aaai.org/index.php/AAAI-SS/article/view/27493, can help readers explore the topic of bias in ML more deeply.

[Answer: The articles have added and commented.]

7. Authors need to highlight their unique contributions in the very beginning.

[Answer: The contributions are listed in the Introduction part in the shape of claims.]

Comments on the Quality of English Language

can be improved, break longer paras into smallers 

[Answer: The text has been edited by avoiding long sentences.]

Round 2

Reviewer 1 Report

Comments and Suggestions for Authors

The authors revised the paper to my satisfaction. I have no further comments.

Reviewer 2 Report

Comments and Suggestions for Authors

Most of the comments have been addressed. Thank you

Reviewer 4 Report

Comments and Suggestions for Authors

Most comments are addressed 

Comments on the Quality of English Language

Fine